# Mechanistic insights into allosteric regulation of the A$_{2A}$ adenosine G protein-coupled receptor by physiological cations

Libin Ye[1,2], Chris Neale[3], Adnan Sljoka[4], Brent Lyda[5], Dmitry Pichugin[1,2], Nobuyuki Tsuchimura[4], Sacha T. Larda[1,2], Régis Pomès [2,6], Angel E. García[3], Oliver P. Ernst [2,7], Roger K. Sunahara [5] & R. Scott Prosser[1,2]

Cations play key roles in regulating G-protein-coupled receptors (GPCRs), although their mechanisms are poorly understood. Here, $^{19}$F NMR is used to delineate the effects of cations on functional states of the adenosine A$_{2A}$ GPCR. While Na$^+$ reinforces an inactive ensemble and a partial-agonist stabilized state, Ca$^{2+}$ and Mg$^{2+}$ shift the equilibrium toward active states. Positive allosteric effects of divalent cations are more pronounced with agonist and a G-protein-derived peptide. In cell membranes, divalent cations enhance both the affinity and fraction of the high affinity agonist-bound state. Molecular dynamics simulations suggest high concentrations of divalent cations bridge specific extracellular acidic residues, bringing TM5 and TM6 together at the extracellular surface and allosterically driving open the G-protein-binding cleft as shown by rigidity-transmission allostery theory. An understanding of cation allostery should enable the design of allosteric agents and enhance our understanding of GPCR regulation in the cellular milieu.

[1] Department of Chemistry, University of Toronto, 3359 Mississauga Road North, Mississauga, ON L5L 1C6, Canada. [2] Department of Biochemistry, University of Toronto, 1 King's College Circle, Toronto, ON M5S 1A8, Canada. [3] Center for Nonlinear Studies, Los Alamos National Laboratory, Los Alamos, NM 87545, USA. [4] Department of Informatics, School of Science and Technology, CREST, Japan Science and Technology Agency (JST), Kwansei Gakuin University, Nishinomiya 530-0012, Japan. [5] Department of Pharmacology, University of California San Diego School of Medicine, 9500 Gilman Drive, La Jolla, CA 92093, USA. [6] Molecular Structure and Function, The Hospital for Sick Children, 686 University Avenue, Toronto, ON M5G OA4, Canada. [7] Department of Molecular Genetics, University of Toronto, 1 King's College Circle, Toronto, ON M5S 1A8, Canada. Correspondence and requests for materials should be addressed to R.S.P. (email: scott.prosser@utoronto.ca)

Humans have over 800 G-protein-coupled receptors (GPCRs), whose capacities to accomplish signaling through interaction with intracellular partners are regulated by extracellular ligands. GPCRs are by no means simple ligand-activated on/off switches and are best represented in terms of an ensemble of multiple inactive, intermediate, and active receptor states[1–3]. The relative populations of these states are influenced by many cell constituents, including physiological cations[4–6]. For instance, early ligand binding studies of opiate receptors in rat brain homogenate showed that sodium increased antagonist affinity while decreasing agonist affinity[7]. In contrast, divalent $Mg^{2+}$, $Ca^{2+}$, and $Mn^{2+}$ enhanced binding of $\mu$-, $\delta$-, and $\kappa$-opioid receptor agonists in brain homogenates[8,9]. Allosteric effects of $Mg^{2+}$, $Ca^{2+}$, and $Na^+$ have been observed with $A_{2A}R$[10–12] and the muscarinic $M_2$ receptor[13], while $Ca^{2+}$ is known to allosterically regulate many family C GPCRs[14–16]. Most recently, high-resolution crystallographic studies of $A_{2A}R$, the $\delta$-opioid receptor, PAR1 protease activated receptor, and the $\beta_1$-adrenergic receptor show that $Na^+$ plays an integral role in stabilizing a functionally inactive conformational state[17–20]. In particular, the 1.8 Å X-ray crystal structure of $A_{2A}R$ (PDB ID: 4EIY) revealed an $Na^+$ pocket, coordinated by water molecules and several highly conserved residues, including $D52^{2.50}$, $S91^{3.39}$, $W246^{6.48}$, and $N284^{7.49}$ (superscripts refer to the Ballesteros and Weinstein nomenclature[21]). The charged groups lining this pocket interface with an almost contiguous network of water molecules spanning the transmembrane domain and linking transmembrane helices (TM) 1, 2, 3, 6, and 7 in the inactive state. Remarkably, the sodium pocket appears to be a common facet of most non-olfactory class A receptors[22,23], although much less is known about binding sites associated with divalent cations and their underlying allosteric mechanisms.

Due to the exceptional sensitivity of the fluorine nuclear spin to electrostatic environment, [19]F NMR has been used to quantitatively monitor conformational equilibria in GPCRs[1–3,24]. Using a next-generation thiol-specific trifluoromethyl tag, this methodology was applied to $A_{2A}R$, revealing an ensemble of inactive states in which a stabilizing ionic lock between $R102^{3.50}$ and $E228^{6.30}$ flickers on a millisecond timescale between an on-state $(S_1)$ and off-state $(S_2)$[2], in slow exchange with two distinct activation intermediates, $S_3$ and $S_{3'}$[1]. Interestingly, the relative populations of $S_3$ and $S_{3'}$ were increased by partial and full agonist, respectively[1]. The activation process can be viewed from the perspective of a free energy landscape, wherein ligands influence both the populations and lifetimes of key functional states. In the current study, we make use of the identical truncated version of human $A_{2A}R$ (2–317) containing a single cysteine mutation $(V229C^{6.31})$, which was labeled with a trifluoromethyl moiety (BTFMA)[25]. Upon reconstituting $A_{2A}R$ in MNG-3 detergent micelles, [19]F NMR revealed the effect of the mono- and divalent cations on the balance of states, and their interplay with orthosteric ligands and a peptide derived from the C-terminal domain of the α subunit of the stimulatory G protein (i.e. $G\alpha_s$ peptide). Traditional radioligand binding assays of the identical construct either by itself or co-expressed with G protein also revealed the allosteric effects of these cations in insect cell membranes. Interaction dynamics with cations are also key to understanding cation egress potentially associated with the activation process. This is demonstrated by [25]Mg NMR and [23]Na NMR CPMG relaxation dispersion measurements, which in the latter case provide a measure of the bound state lifetime of the sodium ion for apo- $A_{2A}R$ and upon saturation with inverse agonist. Explicit-solvent all atom molecular dynamics (MD) simulations of $A_{2A}R$ in a lipid bilayer allow us to predict likely divalent cation binding sites in the receptor and validate interaction dynamics. Subsequent divalent cation bound structures

can then be analyzed using rigidity-transmission allostery (RTA) algorithms that assess the extent to which cation binding allosterically influences G protein binding.

## Results

**Negative allosteric modulation by sodium.** While physiological sodium cation concentrations are around 140 mM, [19]F NMR spectra of MNG-reconstituted $A_{2A}R$ were recorded over a broad range of sodium concentrations to ascertain the relative effect of sodium on the conformational ensemble. Figure 1a, c reveals that the addition of sodium to apo $A_{2A}R$ shifts the equilibrium toward the inactive ensemble $S_{1–2}$ and the $S_3$ state, at the expense of $S_{3'}$. Here, spectra are deconvolved into three resonances, referred to as $S_{1–2}$, $S_3$, and $S_{3'}$, in keeping with previously published work[1]. As discussed previously, $T_2$ relaxation experiments were performed to estimate linewidths that proved key to obtaining robust deconvolutions.[1] The observation that added sodium increases the fraction of conformers associated with the inactive ensemble is consistent with prior radioligand binding studies in rat striatal membranes, which showed sodium slowed the off-rate of an antagonist[26]. Thus far, several crystal structures (Fig. 1b) corroborate the observation of a conserved bound sodium ion that associated exclusively with the inactive state[17,20,22]. Moreover, many class A receptors are believed to bind sodium through a conserved pocket identified by amino acid alignments (Fig. 1d)[22]. The observation that sodium also increases the $S_3$ population at higher sodium concentrations may imply a second weaker affinity binding site associated with this activation intermediate or that the sodium-binding pocket, identified by crystallography, is still accessible to sodium in this activation intermediate. We note that control experiments in which KCl was titrated in the presence of 10 mM or 100 mM NaCl revealed no significant change in the [19]F NMR spectra (Supplementary Fig. 1), in keeping with previous radioligand studies of $A_{2A}R$ showing a reduced degree of allosteric effects by potassium[27] or lithium[11] salts.

**Dynamics of $Na^+$ binding to $A_{2A}R$ revealed by [23]Na NMR.** An [23]Na NMR binding isotherm (Fig. 2a, b) reveals that sodium undergoes rapid exchange between free and receptor bound states, with a dissociation constant of 60 mM to apo $A_{2A}R$. Amiloride analogs are also potent allosteric compounds suggested to bind to a region that overlaps the sodium-binding pocket[22,26,28]. In keeping with earlier suggestions that sodium and amiloride analogs act at a common allosteric site[26], the addition of an amiloride analog, 5-(N,N-hexamethylene)amiloride (HMA) was shown to compete with sodium as evidenced by the weakened response in both line broadening and chemical shift perturbations in the [23]Na NMR binding isotherm (Fig. 2c). Moreover a docking study (Fig. 2d) also suggests that sodium and HMA binding to apo $A_{2A}R$ occur at the primary sodium-binding site defined by Liu et al.[17]. Relaxation dispersions associated with [23]Na CPMG experiments, shown in Fig. 2e, provide an additional measure of cation binding and exchange dynamics. Assuming that sodium adopts a simple equilibrium between a bound- and a free-state, it is possible to extend the above binding isotherms with [23]Na CPMG measurements, which provide a measure of the bound state lifetime. Via a global fit of [23]Na relaxation dispersion profiles at two magnetic field strengths, the addition of inverse agonist, ZM241385, increases the bound fraction by 20% and the average bound state lifetime from 480 μs to 630 μs.

**Positive allosteric modulation by divalent cations.** In contrast to $Na^+$, increasing amounts of $Ca^{2+}$ or $Mg^{2+}$ shift the equilibrium of apo $A_{2A}R$ toward the active states, $S_3$ and $S_{3'}$, as shown in

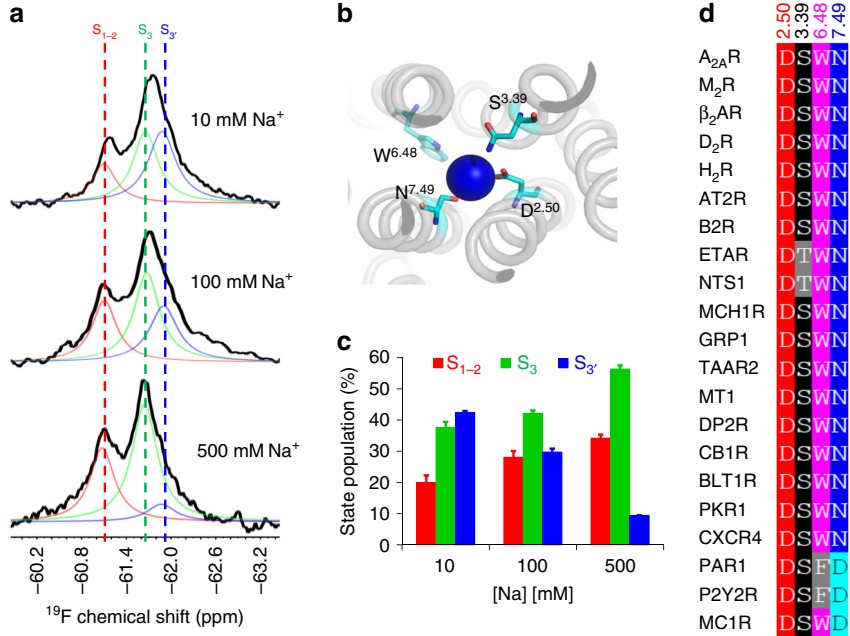

**Fig. 1** Allosteric modulation of $A_{2A}R$ by the monovalent cation $Na^+$. **a** $^{19}F$ NMR spectra of 200 μM BTFMA-labeled $A_{2A}R$-V229C as a function of $Na^+$ concentration (10 mM, 100 mM and 500 mM NaCl). Three distinct resonances correspond to inactive ($S_{1–2}$, shown in red) and active states ($S_3$ and $S_{3'}$, shown in green and blue, respectively). **b** Top view showing conserved residues ($D^{2.50}$, $S^{3.39}$, $W^{6.48}$ and $N^{7.49}$) associated with the sodium pocket of $A_{2A}R$. **c** Population histogram of states, $S_{1–2}$, $S_3$ and $S_{3'}$, corresponding to spectra shown in (**a**). **d** Alignment of conserved residues associated with the sodium pocket in class A GPCRs. Error bars are defined based on the difference between the fitted area associated with the deconvolved spectrum and the area associated with the experimental spectrum. This error bar definition was applied into all spectra without specific description

Fig. 3a and Supplementary Fig. 2, and corresponding histograms (Fig. 3d, e). However, these effects are observed only at very high cation concentrations (i.e. between 100 mM and 500 mM). In contrast, the response to divalent cation appears to be much stronger in the presence of full agonist N-ethyl-5′-carboxamido adenosine (NECA) and $G\alpha_s$ peptide (i.e. the C-terminal helix from $G\alpha_s$, known to bind to $A_{2A}R$[29]). In this case, the addition of 20 mM $Ca^{2+}$ or $Mg^{2+}$ gives rise to a detectable increase in the proportion of active states, and in particular, $S_{3'}$ (Fig. 3b–e and Supplementary Figs. 2b, c). In contrast, this cooperative enhancement in the active ensemble fraction is not observed when $Ca^{2+}$ or $Mg^{2+}$ are combined with partial agonist. While both $Ca^{2+}$ and $Mg^{2+}$ clearly shift equilibria toward the active states, their effect on G-protein-binding affinity can be directly assessed by $^{19}F$ NMR of the $G\alpha_s$ peptide, which in this case was tagged by the trifluoromethyl moiety described above. Building on prior studies[1], $^{19}F$ NMR spectra of labeled peptide (Supplementary Fig. 3b) show clear line broadening with the addition of apo $A_{2A}R$ (Supplementary Fig. 3a). Moreover, the spectra can be deconvolved into two components, one of which is enhanced through the addition of full agonist (NECA). The combination of agonist and either $Ca^{2+}$ or $Mg^{2+}$ brings about an even greater enhancement of this second component, which we ascribe to the $G\alpha_s$ peptide when bound to receptor (Supplementary Figs. 3a and 3c).

**Cooperativity recapitulated by radioligand binding assays.** The above $^{19}F$ NMR observations are further recapitulated in radioligand binding assays in *Sf9* cell membranes in which the identical $A_{2A}R$ construct was co-expressed with the heterotrimeric stimulatory G protein, $G_{\alpha s\beta\gamma}$. Both $Mg^{2+}$ and $Ca^{2+}$ are shown to significantly enhance $[^3H]NECA$ (10 nM) binding with half-maximal cation concentrations of 350–400 μM while $Zn^{2+}$, by contrast, is observed to reduce high affinity binding.

Additionally, $Mg^{2+}$ and $Ca^{2+}$ enhance maximal specific $[^3H]$ NECA binding and affinity ($K_d$) as observed in high affinity saturation experiments (Fig. 4a, b, respectively). Similar allosteric effects of $Ca^{2+}$ and $Mg^{2+}$ on agonist binding, $[^3H]CGS-21680$, were observed in studies of $A_{2A}R$ in rat striatal membranes[11]. While no effect was observed on NECA inhibition of $[^3H]$ ZM241385 binding in membranes expressing $A_{2A}R$ alone, divalent cations enhanced both the affinity ($K_{high}$) and fraction of high affinity NECA states when G proteins are co-expressed with receptor (Fig. 4c, d and Table 1) suggesting that the positive allostery promoted by $Ca^{2+}$ and $Mg^{2+}$ is exclusive to a high affinity state or some activation intermediate state of $A_{2A}R$.

**Molecular dynamics simulations for divalent cation binding.** To identify potential binding sites for divalent cations, all-atom MD simulations were performed on active or active-like states of apo $A_{2A}R$, in the presence of a large excess of cations, which were initially randomly distributed. Figures 5 and 6 depict the overall spatial distribution of cations associated with the receptor, based on fifteen 1 μs simulations, initiated with crystal structures of either of three active states determined recently by X-ray crystallography[27,30,31]. In these simulations, cations frequently bound to a subset of preferred residues, engaging with polar and (more commonly) acidic residues, while in some cases becoming partially desolvated (Supplemental Fig. 4). Stable cation-bound complexes identified by these MD simulations are shown in Fig. 5 and summarized in Supplementary Table 1. Monovalent and divalent cations exhibited significantly different binding patterns. While bound sodium is not present in the active state crystal structure of $A_{2A}R$ bound to an engineered G protein (PDB ID: 5G53), MD simulations predict sodium nevertheless samples a region in the vicinity of residues known to form the sodium pocket in the inactive state. Generally, binding of divalent cations to either of two agonist-bound $A_{2A}R$ models (PDB IDs: 2YDO

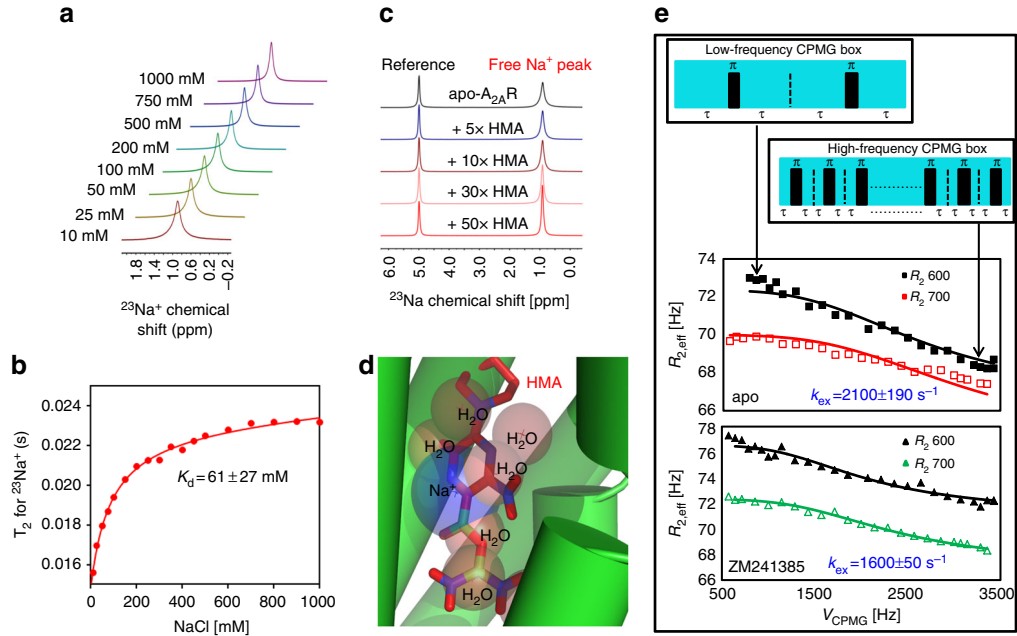

**Fig. 2** Effect of the allosteric modulator HMA on the sodium pocket. **a** $^{23}Na$ NMR spectra as a function of $Na^+$ concentration (10–1000 mM NaCl) in the presence of 50 μM apo $A_{2A}R$-V229C. **b** Line width-based binding isotherm for NaCl derived from (**a**), showing a dissociation constant of $K_d$, of 61 ± 27 mM. (The deviation was defined by three replicates and was applied into all ±values unless described otherwise.) **c** $^{23}Na$ NMR spectra of $A_{2A}R$-V229C, in the presence of 100 mM NaCl, as a function of stoichiometric equivalents of the allosteric modulator HMA. **d** Low-energy pose of HMA docked to $A_{2A}R$ in the absence of sodium, showing sodium (purple) and water molecules (red) from PDB ID: 4EIY, overlaid with HMA, which was docked in the absence of sodium and internal water molecules. **e** $^{23}Na$ NMR CPMG relaxation series of apo $A_{2A}R$-V229C. Each spectrum was acquired using 256 scans with a constant $T_2$-refocusing period of 70 ms and a pulse refocusing frequency, $V_{CPMG}$, as described in the figure. Note that no discernible dispersion was observed with the identical experiment performed in the presence of MNG-3 detergent micelles without $A_{2A}R$

and 3QAK) was so similar that the results were averaged. All of the divalent cation-mediated interactions described below persisted for the duration of the simulation and are classified in terms of: (i) bridging between acidic residues, as in D170-D261 (Fig. 5b), E151-E161 (Fig. 5c), E151-E169-D170 (Fig. 5d), and E169-D170 (Fig. 5e), (ii) cross-linking between acidic residues and surrounding lipid phosphoryl groups, as in the case of E169, E219, E228, D261, E312, and E212 (Fig. 5g and Supplementary Table 1), (iii) localization in the vicinity of the orthosteric site and the sodium-binding pocket, dynamically overlapping both regions and interacting with acidic residues $E13^{1.39}$ and $D52^{2.50}$ (Fig. 5f), and (iv) inhibition of the ionic lock (i.e., $DR^{3.50}Y$—$E228^{6.30}$) through interaction with $E228^{6.30}$ (Fig. 5h).

**Allosteric propagation of divalent cation bridging**. Using the above MD-derived structures in complex with divalent cations, it is possible to evaluate their roles in allosteric transmission using rigidity transmission allostery (RTA) algorithms, which are based on an extension of the program Floppy Inclusion and Rigid Substructure Topography (FIRST)[32–34]. The receptor is first decomposed into rigid clusters and flexible regions and RTA algorithms are used to estimate the number of non-trivial degrees of freedom in the system. Binding at one site introduces new local constraints and thus, a perturbation in the degrees of freedom. Allosteric transmission constitutes a change in degrees of freedom at the G-protein-binding domain resulting from an initial perturbation. Mechanically, a change in conformation induced by a binding ligand at one site may result in a change in shape or conformation at a second site. The extent of coupling is evaluated in terms of the presence of rigidity-transmission (change in degrees of freedom) between these two sites. To explore this, we sequentially perturbed the rigidity of a window of three

consecutive residues ($r, r + 1, r + 2$) associated with extracellular residues within ECL1, ECL2, and ECL3, and calculated the degrees of freedom that could be transmitted from the extracellular domain to the G-protein-binding region, defined by residues 208, 230, and 291 at the ends of TM5, 6, and 7 (Fig. 7a). Residues were labeled as allosteric hot spots based on the intensity of transmission of degrees of freedom. This exercise was repeated for conformers generated from MD simulations and represented in Fig. 5b–e. As shown in Fig. 7b, using the active state crystal structure (PDB ID: 2YDO), rigidifying divalent cation bridges E151-E161, E151-E169-D170, and D261-170 significantly increased the intensity of degrees of freedom transmission (DOF), suggesting that restrictions imposed by cation bridges are allosterically propagated in the form of rigidity-transmission to the G-protein-binding region.

## Discussion

In the current study, we were able to recapitulate prior observations of negative allosteric effects of sodium, albeit in terms of direct observations of changes in state populations. Here we also observe an enhancement of the fraction of both inactive ($S_{1–2}$) and active ($S_3$) conformers over a modest range of sodium chloride concentrations, at the expense of the fully active state, $S_{3'}$. Of particular interest, the population ratio of $S_3$ to $S_{12}$ conformers increases at much higher sodium concentrations, suggestive of a second lower affinity $Na^+$ binding site that might further stabilize the $S_3$ activation intermediate. Recent $^{23}Na$ NMR binding isotherms of $A_{2A}R$, performed in the presence of saturating amounts of the agonist (NECA) also reveal the presence of a sodium-binding site, despite the fact that there is no substantial inactive conformer (unpublished results). Given that X-ray crystal structures of NECA-stabilized $A_{2A}R$ do not show evidence for

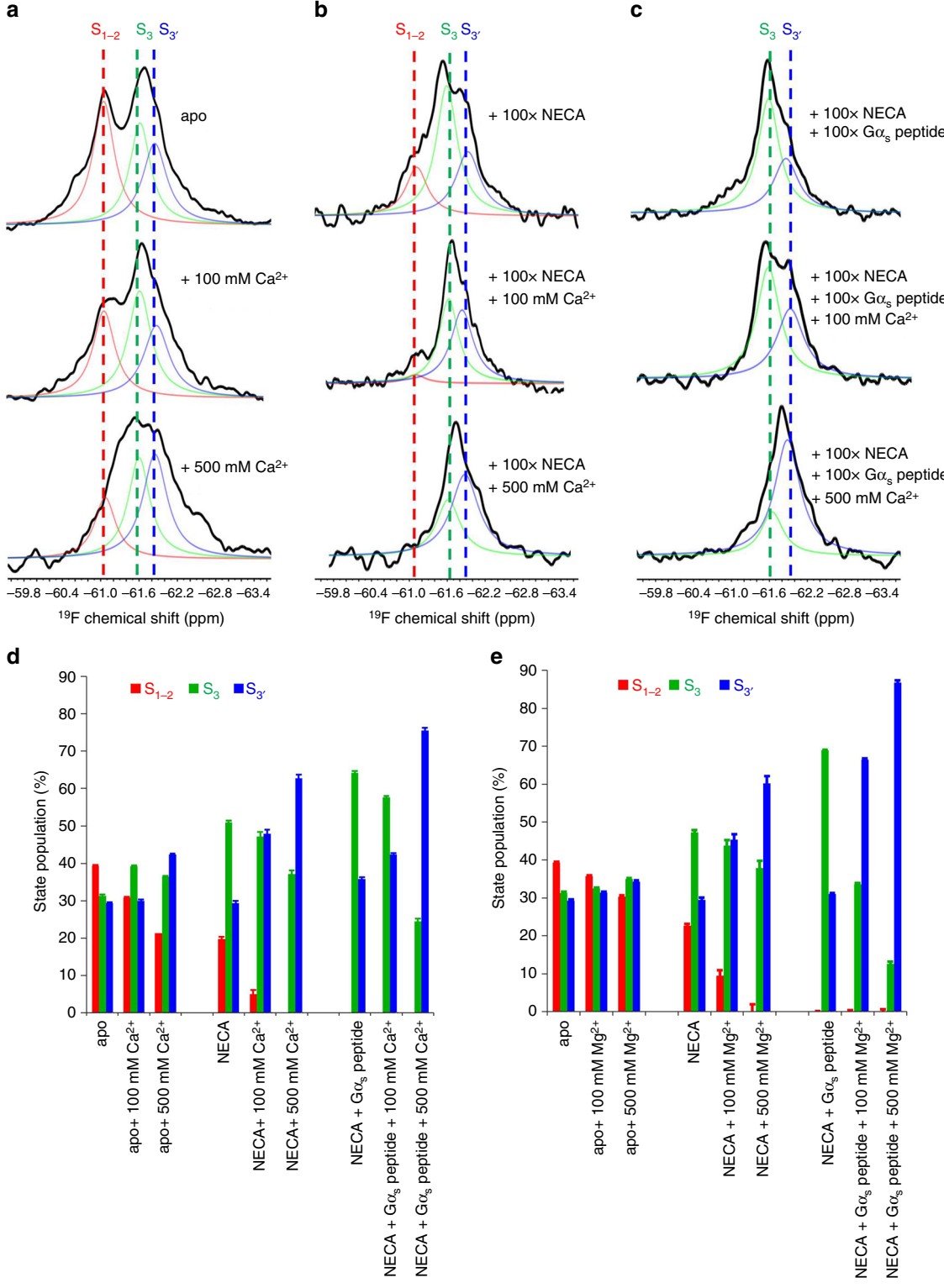

**Fig. 3** Comparison of the effects of divalent cations $Ca^{2+}$ and $Mg^{2+}$ on the receptor. **a** $^{19}F$ NMR spectra of 100 μM apo BTFMA-labeled $A_{2A}R$-V229C as a function of $Ca^{2+}$ concentration (0, 100, and 500 mM $CaCl_2$). **b** $^{19}F$ NMR spectra of 100 μM NECA-saturated BTFMA-labeled $A_{2A}R$-V229C as a function of $Ca^{2+}$ concentration (0, 100, and 500 mM $CaCl_2$). **c** $^{19}F$ NMR spectra of both NECA- and $G\alpha_s$ peptide-saturated BTFMA-labeled $A_{2A}R$-V229C as a function of $Ca^{2+}$ concentration (0, 100, and 500 mM $CaCl_2$). **d** Population histogram of states, $S_{1-2}$, $S_3$ and $S_{3'}$, upon titration of different concentrations of $CaCl_2$, in combination with saturating NECA and $G\alpha_s$ peptide. **e** Population histogram of states, $S_{1-2}$, $S_3$ and $S_{3'}$, upon titration of different concentrations of $MgCl_2$, in combination with saturating NECA and $G\alpha_s$ peptide (refer to Supplementary Fig. 2 for corresponding $^{19}F$ NMR spectra). NECA and $G\alpha_s$ peptide were used in 100× excess over the receptor. All samples contained 100 mM $Na^+$

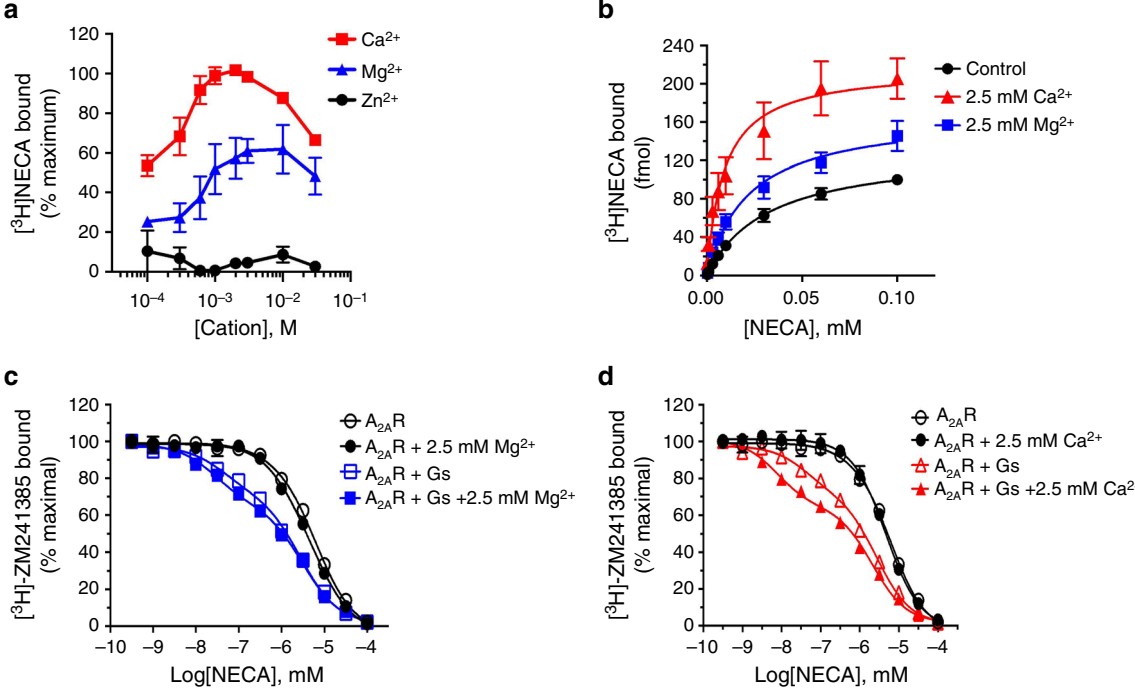

**Fig. 4** Radioligand binding assay in $A_{2A}R/G_s$-infected *Sf*9 membranes. **a** Cations, $Mg^{2+}$ or $Ca^{2+}$, enhance specific [$^3$H]NECA binding in $A_{2A}/G_s$-infected *Sf*9 membranes, while $Zn^{2+}$ displayed reduced binding. **b** Specific [$^3$H]NECA high affinity saturations ±2.5 mM $Ca^{2+}$ or ±2.5 mM $Mg^{2+}$. **c** Divalent cation, $Mg^{2+}$, enhances the affinity of NECA binding and the fraction of high affinity binding sites on $A_{2A}R$ in a G-protein-dependent manner. **d** Divalent cation, $Ca^{2+}$, enhances the affinity of NECA binding and the fraction of high affinity binding sites on $A_{2A}R$ in a G-protein-dependent manner. Error bars for radioligand binding assays are defined based on three replicates

---

**Table 1 Summary of effects of divalent cations on NECA inhibition of [$^3$H]ZM241385**

| Membranes | Cation | $K_{high}$ (nM)[b] | $K_{low}$ (nM)[b] | Fraction of $K_{high}$ (%)[b] |
|---|---|---|---|---|
| $A_{2A}R$ alone[a] | Control | | 1781.00 ± 175.00 | |
| | $Mg^{2+}$ | | 1332.00 ± 82.00 | |
| | $Ca^{2+}$ | | 1533.00 ± 146.00 | |
| $A_{2A}R$ + Gs | Control | 14.10 ± 0.70 | 950.00 ± 97.00 | 26.60 ± 2.30 |
| | $Mg^{2+}$ | 9.18 ± 1.87 | 1060.00 ± 124.00 | 33.70 ± 2.03 |
| | $Ca^{2+}$ | 2.58 ± 0.58 | 708.00 ± 92.00 | 36.70 ± 1.88 |

[a] Data for $A_{2A}R$ alone were fit to a single site
[b] $K_{high}$, $K_{low}$ and fraction of $K_{high}$ were fitted using non-linear regression using Prism software (GraphPad, La Jolla, CA)

---

bound sodium, it is therefore likely that the activation intermediate, $S_3$, known to be stabilized by partial agonist, can also bind $Na^+$.

Sodium interactions are observed to be transient and dynamic, based on $^{23}$Na CPMG relaxation dispersion measurements at multiple fields. In this study, the average residency time of a sodium ion in apo $A_{2A}R$ is found to be 480 μs. As sodium is part of an extensive allosteric water network in the inactive state, its release from the primary binding site in apo $A_{2A}R$ may also allow for transient excursions to active conformations, thereby facilitating basal activity in class A receptors. Normally, the sodium-binding site would be expected to be rapidly replenished by the extracellular pool of sodium ions. However, binding by agonist would block such access and bias egress of sodium toward the intracellular milieu (i.e. down the sodium concentration gradient), potentially affording free energy to the activation process as has been elegantly hypothesized.[22] The observation that the activation intermediate, $S_3$, is also stabilized by sodium corroborates the idea that sodium dynamics and its eventual cytosolic egress is intrinsically involved in the activation process and that

intermediate binding sites may be involved in the egress of sodium[35].

$^{19}$F NMR and $^{23}$Na NMR results suggest that the addition of inverse agonist both increases the inactive conformer population and the average $Na^+$-bound state lifetime (i.e. 630 μs), thereby curtailing excursions to activation intermediates or sampling of other conformers. Prior studies have shown that the addition of inverse agonist also leaves comparable populations of both the inactive ($S_{1-2}$) and activation intermediate ($S_3$) states. Consequently, the above average $Na^+$-bound state lifetime will represent a weighted average between each of these two states and it may be the case that the $Na^+$-bound-state lifetime in the ZM-stabilized inactive conformer is much longer. Note that the fits of the $^{23}$Na CPMG dispersions to the apo $A_{2A}R$ data set are poorer than those in the presence of inverse agonist. This likely relates to the fact that the apo receptor consists of a broad ensemble of both inactive and active-like conformers, whose affinities interaction dynamics with sodium are quite different.

The positive allosteric effects associated with $Ca^{2+}$ and $Mg^{2+}$ on $A_{2A}R$ are also recapitulated in the $^{19}$F NMR spectra, albeit at

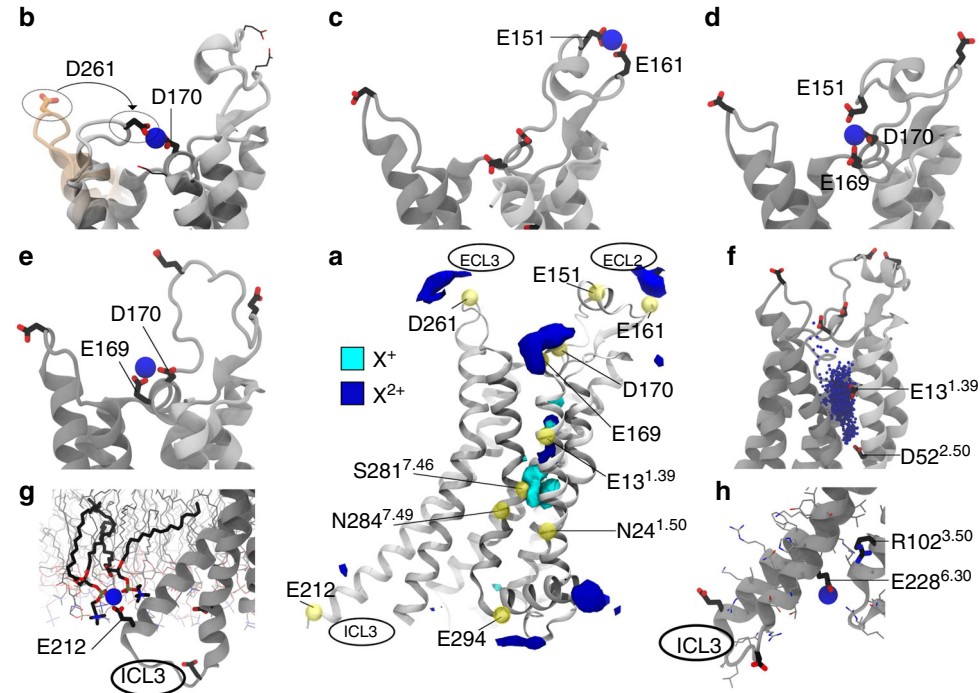

**Fig. 5** Atomistic MD simulations of divalent cation binding by A$_{2A}$R. **a** Spatial distribution of cations around A$_{2A}$R. Surfaces enclose cation density 3 k$_B$T above homogeneous populations for monovalent (cyan) and divalent (blue) cations in simulations of three agonist-bound A$_{2A}$R crystal structures [PDB IDs: 2YDO, 3QAK and 5G53]. **b–e** Divalent cation bridges observed in the receptor's extracellular domain. Orange ECL3 in part (**b**) shows the initial conformation and highlights the reorganization of D261 during MD simulation. **f** Divalent cation binding close to the sodium pocket, coordinated by acidic residues E13$^{1.39}$ or D52$^{2.50}$. **g** Divalent cation bridging between E212 and a phosphoryl group in the lipid bilayer. **h** Divalent cation binding to E228$^{6.30}$ (ionic lock: DR102$^{3.50}$Y—E228$^{6.30}$) (see also Fig. 7)

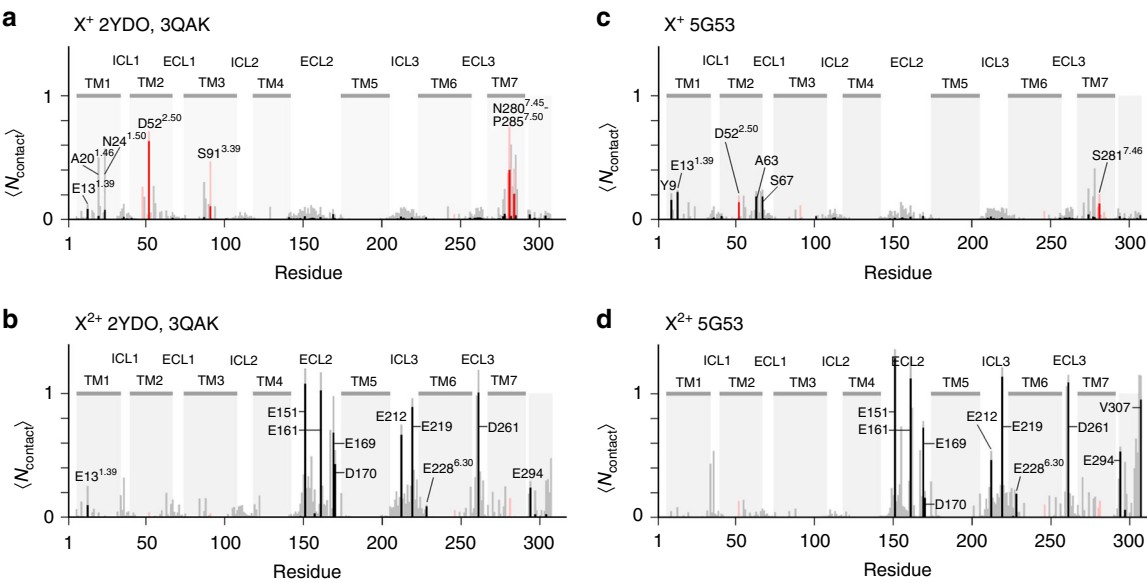

**Fig. 6** Cation interactions with A$_{2A}$R. Histograms show the proportion of ions within 0.6 nm (light bars) or 0.3 nm (dark bars) of any receptor atom, $\langle N_{\text{contact}} \rangle$, as a function of residue number averaged over (**a, b**) five 1-μs simulations from each of two agonist-bound A$_{2A}$R crystal structures [PDB IDs: 2YDO and 3QAK], and (**c, d**) five 1-μs simulations from the agonist and G-protein-bound A$_{2A}$R crystal structure [PDB ID: 5G53]. Contacts are shown for monovalent (**a, c**) and divalent cations (**b, d**). Note that red bars designate residues known to coordinate Na$^+$ in the inactive state. Transmembrane (TM) helices, intracellular loops (ICL), and extracellular loops (ECL) are indicated

much higher concentrations. We also note that the receptor concentrations used in the NMR studies are orders of magnitude higher than those used in the radioligand studies. In insect cells, we observed pronounced enhancement in NECA binding with Mg$^{2+}$ and Ca$^{2+}$ concentrations of 350–400 μM, as discussed

above. Using $^{19}$F NMR in detergent-reconstituted receptors, we observe clear enhancement in the fraction of active state conformers in the presence of saturating amounts of NECA and the C-terminal Gα$_s$ peptide, using Mg$^{2+}$ concentrations of 20 mM, as shown in Supplementary Figure 5. This discrepancy in cation

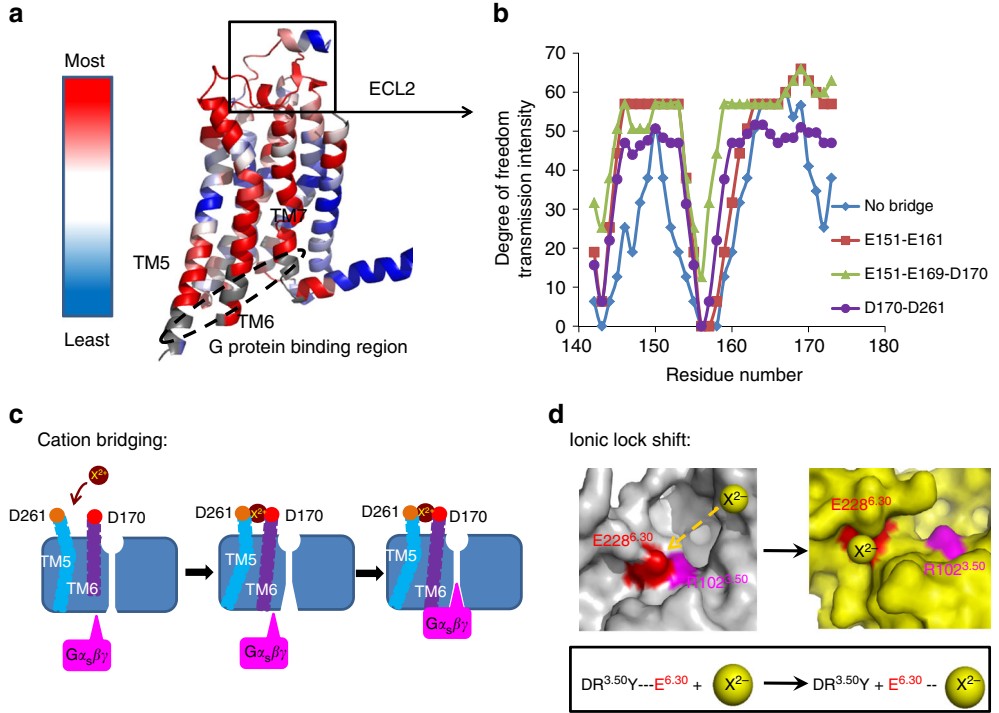

**Fig. 7** Proposed mechanism of allosteric activation of A$_{2A}$R by divalent cations. **a** Using the RTA analysis, regions are colored based on the intensity of degrees of freedom (DOF) transmitted to the G-protein-binding region (depicted in a dashed ellipse) with red regions having highest transmission. **b** Comparison of intensity of transmission of degrees of freedom to the G-protein-binding site as a result of perturbing rigidity through rigidification of sliding windows of three consecutive windows in the region corresponding to ECL2, with and without cation bridges ascertained by MD simulations. **c** Cartoon representation for divalent cation bridge of D261-D170 affecting A$_{2A}$R activation. **d** Potential mechanism of disruption of the ionic lock by divalent cation binding

concentration between the NMR results and the radioligand measurements may arise from the use of detergents to reconstitute the receptor or possibly, a cooperative interaction between G protein, Mg$^{2+}$, and receptor in cell membranes, not recapitulated with the G$\alpha_s$ peptide used in the NMR experiments.

In the absence of high resolution X-ray crystal structures, our understanding of the mechanism of positive allostery through binding by divalent cations is limited. However, the all atom MD simulations of A$_{2A}$R in the presence of cations provide several likely binding modes and means of positive allostery. Firstly, it cannot be discounted that the divalent cations may bind in the vicinity of the sodium-binding site in the active state, as suggested by Fig. 5f, thereby establishing an allosteric network which may help promote G protein binding. Also, binding of divalent cations to E228$^{6.30}$, which functions prominently in the ionic lock, would clearly serve to screen electrostatic interactions with R102$^{3.50}$, thereby preventing excursions toward an inactive conformer, effectively increasing the active state lifetime (Fig. 5h).

One possible mode of allosteric enhancement is expected to arise from bridging of acidic residues, located in either ECL2 or ECL3, by the divalent cations, as illustrated in Fig. 5a–e. In particular, E151-E161, E151-E169, and E151-D170, located on either end of an extracellular helix within ECL2, and D170-D261 in ECL3 were observed to establish long-lived electrostatic interactions via an intervening divalent ion in the MD simulations. We note that the crystal structure of agonist-bound A$_{2A}$R reveals a putative binding site between E169 and the purine group of the agonist (PDB ID: 2YDO), while the majority of A$_{2A}$R crystal structures show the extracellular helix in ECL2 to be positioned such that the above extracellular inter-residue contacts could not be attained without conformational changes in the protein backbone. However, it should be emphasized that in the active

state crystal structures of thermostabilized A$_{2A}$R (PDB IDs: 2YDO) and at least one member of the unit cell of A$_{2A}$R complexed with G protein (PDB ID: 5G53) the extracellular helix is constrained by crystal packing, while in a second active state crystal structure (PDB ID: 3QAK) or a second unit cell member of A$_{2A}$R with G protein (PDB ID: 5G53), the extracellular helix is disordered, in which case large amplitude conformational excursions of ECL2 would be possible. The MD simulations enable us to sample sub-states that are dynamically accessible from the ground state crystal structure, free of crystal packing effects, and in the presence of cations. While the above cation-bridged interactions were observed to be long-lived by MD standards (i.e. microsecond or longer), even transient bridging of acidic residues by divalent cations would be expected to lower the activation barrier toward sampling of active or intermediate receptor states.

We note that the bridging of D170 and D261 by a divalent cation, identified by MD simulations, brings the extracellular ends of TM5 and TM6 in closer proximity (Fig. 5b), mechanically facilitating the outward rotation of TM6 at the intracellular interface (Fig. 8c–f). Indeed, as shown in Fig. 8a, b, D261-D170 inter-residue distances are clearly shortened in active state X-ray crystal structures, in support of the idea that juxtaposition of the extracellular facing ends of TM5 and TM6 helps to prop open TM6 on the cytosolic face. Thus, in keeping with the above RTA allostery transmission analysis, the close association of TM5 and TM6 through these cation-mediated interactions has direct consequences with regard to driving open the cytosolic G-protein-binding cleft, as illustrated in Fig. 7b, c. Finally, binding of divalent cations to E228$^{6.30}$, which functions prominently in the ionic lock, would clearly disfavor return to the inactive state upon activation, as illustrated in Fig. 7d. In contrast to the above

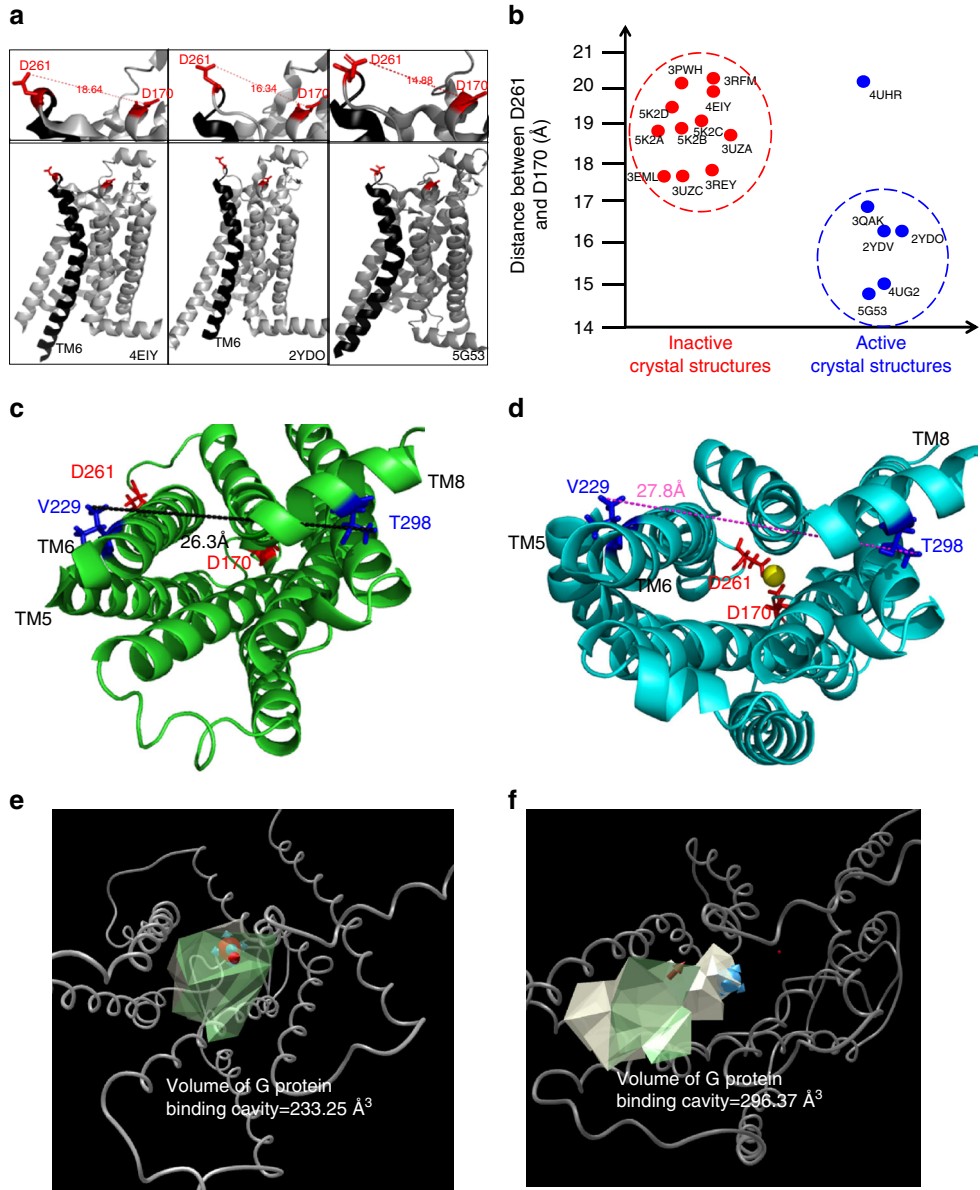

**Fig. 8** Topology of D261-D170 and effects of its bridge on the G-protein-binding cavity. **a** Distances between residues D261 and D170 in crystal structures of inverse agonist (PDB:4EIY) bound-, agonist(PDB:2YDO) bound- and engineered G protein (PDB:5G53) bound-A$_{2A}$R receptors. **b** Distribution of inter-residue distances, D261-D170, from all available A$_{2A}$R crystal structures in the PDB database. Note the above distances refer to those between the most proximal atoms from each residue. **c**, **e** Bottom view of the intracellular G-protein-binding domain of the apo sample revealing the extent of opening of the G-protein-binding cavity through the distance between V229 (TM6) and T298 (TM8) (**c**), and the volume of the cavity (**e**). **d**, **f** Bottom view of the intracellular G-protein-binding domain at the conclusion of the MD trajectory, showing the resultant distance between V229 (TM6) and T298 (TM8) (**d**), and the volume of the G-protein-binding cavity after engagement of the divalent cation (**f**)

mechanism, effects of divalent cation interference with the ionic lock would arise largely from the intracellular pool. Thus, extra- and intracellular concentrations and compartmentalization would be expected to give rise to pronounced local effects of divalent cations.

The role of the extracellular acidic residues in facilitating cation-mediated allostery has been previously considered by Jacobson and colleagues, who focused on radioligand binding to A$_{2A}$R in transfected COS-7[12]. In this case, acidic residues in the second extracellular loop (E151, E161, E169, and D170) were replaced with alanine or glutamine residues[12]. While they were able to recapitulate the positive allostery on the part of Mg$^{2+}$, they were unable to identify a change in the allosteric response based on agonist affinity resulting from either E161A or E169Q

mutations. Full stimulation of adenylyl cyclase was also observed, albeit at thousand-fold higher agonist concentrations. A thorough analysis of the implications in mutating these acidic residues proved problematic given the gross changes in ligand binding seen for most mutations[12]. Moreover, it is also possible that the proposed allosteric effects resulting from bridging key acidic residues arise from multiple transient and cooperative interactions, as proposed in Fig. 5. Given the dual role of the extracellular loops in negotiating ligand access and in cooperativity, it would be difficult to perform a thorough analysis through mutations of extracellular acidic residues. In this sense, the MD studies can theoretically provide essential insight into the role of the extracellular acidic residues in facilitating divalent cation binding and contributing to the activation process.

Bridging of acidic residues by divalent cations appears to be spontaneous and relatively stable from the perspective of MD simulations over 1 μs trajectories, though it is not clear if this is physiologically relevant since very high divalent cation concentrations were used in the MD study. In this case, a conserved divalent ion binding site, resulting from a conformational transition could be missed without an appropriate MD starting structure. However, we note that $^{19}$F NMR studies, which investigated the role of divalent cations in the presence of agonist and $G\alpha_s$ peptide, showed that 20 mM $Mg^{2+}$ resulted in an obvious increase in the active $S_{3'}$ state population whereas $Fe^{2+}$ did not (Supplementary Fig. 6), in keeping with radioligand studies where both $Fe^{2+}$ and $Zn^{2+}$ gave no positive allosteric effects. Assuming $Ca^{2+}$, $Mg^{2+}$, and $Fe^{2+}$ would serve equally well to bridge acidic residues, this could imply that divalent ion facilitated bridging of acidic residues is not the primary allosteric driving force at least at low divalent cation concentrations. Rather, binding of full agonist and the interaction with G protein bring about a unique coordination site that specifically facilitates the observed allosteric effect via $Ca^{2+}$ or $Mg^{2+}$. The results observed by MD studies at high cation concentrations are nonetheless interesting from the perspective of pharmaceutical strategies for positive allosteric modulators (PAMs). MD simulations suggest a compaction of the extracellular domain of the receptor by bridging acidic residues results in a significant opening of the intracellular G-protein-binding cavity volume (Fig. 8e, f) which is corroborated by rigidity theory. Thus, it may be possible to purposefully design a PAM that recapitulates this extracellular compaction, as was recently shown with the allosteric modulator, LY2119620, in studies of the M2 muscarinic acetylcholine receptor[36].

The above results point to an important relationship between cations and GPCR function. First, the sodium-binding site appears to be a common facet of class A GPCRs, and can be traced to certain early microbial rhodopsins[37]. While sodium is a recognized negative allosteric modulator, which predictably stabilizes the inactive ensemble, its interaction with various states of the receptor is observed to be transient, suggesting that sodium may well undergo significant egress in the receptor, possibly playing a role in activation[22] or in facilitating basal excursions to activation intermediates. $^{19}$F NMR provides the means to directly quantify the stabilizing effects of sodium on the conformational ensemble, with the surprising result that the activation intermediate, $S_3$, linked earlier to partial agonism, is also stabilized by sodium. Moreover, $^{23}$Na CPMG measurements provide quantitative measures of cation interaction dynamics in these receptors, and therefore allow us to elaborate on cation-mediated activation mechanisms.

The positive allostery observed with the divalent cations $Ca^{2+}$ and $Mg^{2+}$ stand in contrast to the negative allostery observed with sodium, although all ions appear to stabilize pre-existing states in a manner consistent with conformational selection[1]. MD simulations and rigidity theory calculations point to a possible role of these divalent cations in bridging key acidic residues at high cation concentrations, thereby helping establish an allosteric activation pathway to the G-protein-binding region. These studies suggest that one possible means of achieving positive allosteric modulation is to compact the extracellular regions.

Overall, $^{19}$F NMR reveals pronounced effects of both $Na^+$ and either $Mg^{2+}$ or $Ca^{2+}$ on the population of functional receptor states. Moreover, this is fully recapitulated in radioligand assays in cell membranes at sub-millimolar concentrations. $^{23}$Na (Fig. 2) and $^{25}$Mg NMR (Supplementary Fig. 7) show that these interactions operate on a millisecond timescale, underlining their role in "dynamically" influencing the activation process.

## Methods

**Gene operation and receptor purification.** The construct pPIC9K_F$_\alpha$-Factor-Flag-TEV-A2aARTr316-H10_V229C containing the V229C mutation was generated using a QuikChange Lightning Site-Directed Mutagenesis Kit (Agilent Technologies) according to the manual of the Kit using a single primer[1] of 5′-CCACACTGCAGAAGGAGTGCCATGCTGCCAAGTCAC-3′. Freshly prepared competent cells of the SMD 1163 strain (Δhis4 Δpep4 Δprb1, Invitrogen) of *Pichia pastoris* (ATCC® 28485™; Invitrogen, CA, USA) were electro-transformed with *Pme*I-HF (New England Biolabs) as linearized plasmids by a Gene Pulser II (Bio-Rad). High copy clone selection was performed by a two-stage screening approach established in-house[29,38]. The transformants were first spread on histidine-deficient YNBD plates. After 3–5 days incubation at 30 °C, the resulting colonies were transferred individually onto YPD plates containing 0.25 mg/mL G418 for another incubation of 3–5 days at 30 °C. The colonies were subsequently transferred to YPD plates containing 2 mg/mL G418 for an additional incubation period of 5–7 days and then further transferred onto YPD plates containing 4 mg/mL G418 and incubated for another 5–7 days. One of the surviving colonies was then selected to make competent cells for second-cycle screening. After transformation, the secondary transformants were directly spread onto YPD plates with different G418 concentrations (2 mg/mL and 4 mg/mL), rather than YNBD plates. The colonies that grew on 4 mg/mL G418 YPD plates were further transferred onto 6 mg/mL G418 YPD plates for an additional 5–7-day incubation period. Ten to 15 high-copy colonies picked from YPD plates containing 6 mg/mL G418 were then screened by an immunoblotting assay for further expression.

Reconstitution and functional purification of the high-yield construct was performed as follows[1]: After expression, the cell pellets were washed one time with buffer P1 (50 mM HEPES, pH 7.4) and re-suspended in Lysis buffer P2 (50 mM HEPES, pH 7.4, 100 mM NaCl, 2 mM EDTA, 10% glycerol, 100 U Zymolyase, EDTA-free-proteinase inhibitor) at a ratio of 1:4 (W/V) and left standing for 1 h at room temperature. Yeast cell walls were further disrupted by vortexing for 2 h at 4 °C in the presence of 5 mm glass beads, which were added to the re-suspended cells. Intact cells and cell debris were separated from the membrane suspension by low speed centrifugation (8000 × g, 20 min, 4 °C). The supernatant was collected and centrifuged at 100,000 × g for 1 h and the resulting membrane pellets were then dissolved in 25 mL buffer P3 (50 mM HEPES, pH 7.4, 100 mM NaCl, 2 mM theophylline, 10% glycerol, EDTA-free-protease inhibitor, solution of 1% lauryl maltose neopentyl glycol (MNG) (Anatrace), 0.2% CHS, and 20 mM imidazole) under continuous shaking for about 1–2 h at 4 °C. TALON® Metal Affinity Resin was then added into re-suspended membranes under gentle shaking for 2 h overnight at 4 °C. The receptor conjugated resin was washed extensively with buffer P4 + 1 (50 mM HEPES, pH 7.4, 100 mM NaCl, 4 mM theophylline, 10% glycerol, EDTA-free-protease inhibitor, solution of 0.1% MNG and 0.02% CHS) to remove non-conjugated receptor and impurities in solution. The receptor was eluted from the TALON resin column after addition of 10 column volumes of buffer P5 (50 mM HEPES, pH 7.4, 100 mM NaCl, 10% glycerol, solution of 0.1 % MNG and 0.02% CHS, 250 mM imidazole). The eluant was then concentrated to 5 mL and concentrations of NaCl, imidazole, and theophylline were decreased by washing with buffer P6 (50 mM HEPES, pH 7.4, 0.1% MNG, 0.02% CHS). The receptor was then incubated with XAC-agarose gel for at least 1 h with gentle shaking to allow for the receptor binding to XAC-agarose. The receptor-conjugated XAC-agarose was packed on a disposable column. The column was washed with buffer P7 (50 mM HEPES, pH 7.4, 0.1% MNG, 0.02% CHS, 100 mM NaCl) to remove non-bound receptor. The receptor was then eluted with buffer P8 (50 mM HEPES, pH 7.4, 0.1% MNG, 0.02% CHS, 100 mM NaCl, 20 mM theophylline). The eluant was then concentrated into 1–2 mL and dialyzed extensively against buffer P7 (50 mM HEPES, pH 7.4, 0.1% MNG, 0.02% CHS, 100 mM NaCl) prior to NMR experiments.

**Reagents and biochemical assay.** Receptor concentrations were measured via a Pierce BCA Protein Assay Kit (Thermo Scientific, Rockford, USA) using bovine serum albumin as a standard. Agonist 5′-(N-ethylcarboxamido) adenosine (NECA) was purchased from Sigma-Aldrich. 2-bromo-N-[4-(trifluoromethyl)phenyl]acetamide (BTFMA) was purchased from Apollo Scientific (Manchester, United Kingdom). $G\alpha_s$ peptides (RVFNDARDIIQRMHLRQYELL)[29] used in the study were purchased from China Peptides (Shanghai, P.R. China), whose purities were verified to be over 95% by HPLC. Labeling of the $G\alpha_s$ peptide was achieved by addition of 10× BTFMA, followed by gentle stirring for 6 h at room temperature. An equivalent aliquot of BTFMA was then added to the solution followed by gentle stirring for an additional 6 h to ensure complete labeling of the peptide. The BTFMA-labeled peptide was centrifuged at 4000 × g for 5 min, and the supernatant was extensively dialyzed against HEPES buffer (pH 7.4). The final concentration of BTFMA-labeled peptide was estimated by $^{19}$F-NMR.

**$^{19}$F NMR experiments.** NMR samples typically consisted of 250 μL volumes of 25–200 μM A$_{2A}$R-V229C in 50 mM HEPES buffer (pH 7.4) and 100 mM NaCl, doped with 10% D$_2$O. The receptor was stabilized in 0.1% MNG-3 and 0.02% CHS. All $^{19}$F NMR experiments were performed on a 600 MHz Varian Inova spectrometer using a cryogenic triple resonance probe, with the high frequency channel tunable to either $^1$H or $^{19}$F. Typically, the 90° excitation pulse was 23 μs, the acquisition time was 200 ms, the spectral width was set to 15 kHz, and the

repetition time was 1 s. Most spectra were acquired with 10,000–50,000 scans depending on sample concentration. Processing typically involved zero filling, and exponential apodization equivalent to 20 Hz line broadening.

### $^{23}Na^{+}$ NMR experiments.
The experiments were performed on a 600 MHz Varian VNMRS spectrometer using a room temperature XH broadband probe (X-nucleus on the inner coil). Additional $^{23}$Na NMR measurements were performed on an Agilent DD2 700-MHz spectrometer using OneNMR direct detect broadband probe. $^{23}Na^{+}$ NMR experiments were performed in the presence of either 50 μM or 25 μM $A_{2A}R$. An external $^{23}$Na reference was prepared by mixing 50 mM NaCl and 50 mM $TmCl_3$ in equal volumes prior to loading into the coaxial insert. The resultant $^{23}$Na reference was conveniently shifted downfield by 4 ppm from the free sodium chloride signal associated with the receptor sample. A typical experimental setup included a 7.5 μs 90° excitation pulse, an acquisition time of 250 ms, a spectral width of 15 kHz, and a repetition time of 1.25 s. Most spectra were acquired with 512 scans and yielded a signal to noise ratio greater than 100. Information regarding binding can be reliably obtained by observing the shift of the free peak and line broadening associated with fast exchange between "free" and "bound" states.

### $^{25}Mg^{2+}$ NMR experiments.
The experiment was performed on an Agilent DD2-600 NMR, equipped with a 10 mm low gamma direct detect probe and low frequency pre-amp. The experimental parameters used were as follows: 20 °C, a Larmor frequency of 36.72 MHz, a spectral width of 2.5 kHz, 1.1 s acquisition time, 0.5 s relaxation time, a 30 μs 90° excitation pulse, and between 128 and 2048 scans depending on the signal-to-noise ratio. For the $K_d$ determination of $Mg^{2+}$ the following protocol was used: A 1 M $MgCl_2$ solution with 10 mM $DyCl_3$ in $D_2O$ was used as a reference and was placed in a 5-mm OD tube. A sample of 1.5 mL of the protein solution was placed in a 10-mm OD NMR tube. The reference tube was thoroughly cleaned and inserted into the 10-mm tube and centered with a plastic cap. Titration volumes of 3–37 μL of 4.64 M $MgCl_2$ were added directly to the 10-mm tube which was then gently mixed using a vortex shaker. The sample was allowed to equilibrate in the magnet for 10 min at the set temperature and the sample was then locked, tuned, gradient shimmed and acquired. To determine $K_i$ associated with $Ca^{2+}$ displacement of $Mg^{2+}$, 1.3–14.7 μL of 5.0 M $CaCl_2$ was added directly to the $Mg^{2+}$ saturated receptor dispersion which was then gently mixed using a vortex shaker. The sample was allowed to equilibrate at the set temperature in the magnet for 10 min and then locked, tuned, gradient shimmed and acquired. The $K_d$ and $K_i$ values were determined, using GraphPad Prism 6 software, by fitting the data to the formula, $y = A \times x/(K_d + x)$, where $x$ represents the cation concentration.

### $A_{2A}R$-Gs protein co-expression and radioligand binding assays.
$Sf$9 cells (ATCC®CRL1711™; Invitrogen, CA, USA) were infected with baculoviruses to $A_{2A}R$ receptor alone (MOI~1) or together with $Gα_s$. For $Gα_s$ coinfections the cells were infected with $A_{2A}R$ (MOI~1.0) with $Gα_s$ and $Gβγ$ viruses (MOI~5.0 and 3.5 respectively), to ensure satisfactory expression of $Gα_s$ and functional coupling. Post infection, cultures were grown/incubated for 48 h followed by centrifugation to pellet the cells. Membranes were prepared by resuspension of $Sf$9 pellets in 30 mL of 10 mM HEPES, 10 mM NaCl, 0.5 mM $MgCl_2$, 0.5 mM EDTA, pH 7.4 (buffer A) and homogenized before centrifugation at $20,000 \times g$ for 20 min. The pellet was again re-suspended in 30 mL of (buffer A), homogenized and centrifuged $20,000 \times g$ for 20 min. The remaining pellet was then re-suspended to ~1.0 mg/mL by dounce homogenization in 10 mM HEPES, 10 mM NaCl, 0.5 mM EDTA, pH 7.4 (buffer B) and flash frozen until further use.

Radioligand binding assays: Radioligand binding assays were performed on a 96-well format in a final volumes of 200 μL in 10 mM HEPES pH 7.4 including 100 mM NaCl. Prior to use, membranes were thawed and diluted to appropriate concentrations in buffer B. Two micrograms of membrane protein/well were incubated with radioligand (either agonist [$^3$H]NECA or inverse agonist [$^3$H]ZM241385) in the absence or presence of divalent cations ($MgCl_2$ or $CaCl_2$). For [$^3$H]NECA saturations, membranes at 6 μg protein/well were incubated with increasing concentrations of radioligand (as indicated) in the presence of 2.5 mM $MgCl_2$ or 2.5 mM $CaCl_2$, or buffer alone. For divalent cation dose−response curves, membranes were incubated with 10 nM [$^3$H]NECA in the presence of increasing concentrations of divalent cations: $MgCl_2$, $CaCl_2$, or $ZnCl_2$. For NECA competitions of [$^3$H]ZM241385, membranes ($A_{2A}R$ alone or $A_{2A}R$ + Gs) were incubated with 2 nM [$^3$H]ZM241385 and increasing concentrations of unlabeled NECA (as indicated) in the absence or presence of 2.5 mM $MgCl_2$ or $CaCl_2$. For $A_{2A}$ membranes data were fit to a single site since no high affinity NECA binding is observed whereas $A_{2A}R$ + Gs membranes were fitted to a two-site model. All data were fitted using Prism software (Graphpad, La Jolla, CA). All incubations were performed for 3 h at 25 °C. Reactions were initiated through the addition of membranes and terminated by rapid filtration on glass fiber 96-well microtiter GF/C filter plates, followed by washing with 3× volume of cold, 4 °C, buffer B.

### MD simulations of cation distributions.
Simulation systems comprised apo $A_{2A}R$ in a hydrated zwitterionic lipid bilayer with $Na^{+}$, $K^{+}$, $Ca^{2+}$, $Mg^{2+}$, and $Zn^{2+}$ at 100 mM each and $Cl^{-}$ at 800 mM. Three sets of five 1-μs simulations were conducted, using an initial crystal structure of $A_{2A}R$ obtained with either the endogenous agonist adenosine (PDB 2YDO)[30] or the synthetic agonist UK-432097 (PDB 3QAK)[31] or the agonist NECA and an engineered G protein (PDB 5G53)[27].

### System construction of MD simulations.
Extracellular ligands, water molecules, and other solvents resolved in crystal structures were removed. Missing backbone atoms were modeled with the program Loopy[42,43]. Missing side chain atoms and side chain reversions to wild-type were modeled with the program SCWRL4[44]. Disulfide bonds (C71-C159, C74-C146, C77-C166, and C259-C262) and all hydrogen atoms were placed with the GROMACS tool pdb2gmx[38], and the receptor was then energy minimized. The simulated receptor was not glycosylated, all titratable residues were in their standard states for pH 7, and the protein backbone termini were zwitterionic.

A unique bilayer conformation was constructed for each simulation. To this end, a configuration of a 1-palmitoyl-2-oleoyl-*sn*-glycero-3-phosphocholine (POPC) bilayer with 200 (2YDO simulations) or 166 (3QAK and 5G53 simulations) lipids per leaflet was obtained from the CHARMM-GUI[45] membrane builder[46], solvated with 80 water molecules per lipid, energy minimized, and simulated for 300 ns. A snapshot was extracted at a random time between 50 and 300 ns, translated by a random distance in the bilayer plane (resetting the periodic unit cell), and (randomly) in some cases rotated by 180° to exchange upper and lower leaflets.

The receptor was oriented for insertion into the lipid bilayer using the program LAMBADA[47], and embedded in the bilayer using 20 cycles of the InflateGRO2 routine[47] with double-precision GROMACS steepest descent energy minimization. During this procedure, 12–17 lipids were removed from each leaflet, allowing the final numbers to be asymmetric. Each system was hydrated, disallowing water in the bilayer's hydrophobic core, and ions were added at random locations at least 0.8 nm away from protein, lipid or any other ions. Following energy minimization, the solvent (including lipid and ions) was relaxed over 30 ns without unduly perturbing the receptor by conducting sequential 5-ns simulations using position restraints on: (a) receptor heavy atoms with force constants of $10^4$ and $10^3$ kJ/mol/$nm^2$, and (b) receptor $C_α$ atoms with force constants of $10^3$, $10^2$, 10, and 1 kJ/mol/$nm^2$. All restraints were then removed for production simulation. Each repeat uses different models of missing receptor atoms.

### Protein sequence used in MD simulations.
Residue identifiers always corresponded to the receptor's full-length wild-type sequence (UNIPROT ID: P29274). In our simulations, the adenosine-derived receptor sequence was Δ(M1-G5) and Δ(A317-S412); the UK-432097-derived receptor sequence was Δ(M1-P2) and Δ(R309-S412); and the NECA-derived receptor sequence was Δ(M1-G5) and Δ(L308-S412).

### Simulation parameters.
Molecular dynamics simulations were conducted with a single-precision compilation of version 4.6.7 of the GROMACS simulation package[38]. Macromolecules were modeled by the CHARMM 22 protein force field[39] with grid-based energy correction maps[40], and the CHARMM 36 lipid force field[41] as implemented in GROMACS[48,49]. The water model used in the simulation was TIP3P[50] with CHARMM modifications[39]. Water molecules were rigidified with SETTLE[51] and bond lengths in protein and lipid were constrained with P-LINCS[52] using sixth-order coupling and a single iteration. Lennard−Jones interactions were evaluated using an atom-based cutoff, gradually switching off the potential energies of interactions between 0.8 and 1.2 nm. Coulomb interactions were calculated using the smooth particle-mesh Ewald (PME) method[53,54] with a Fourier grid spacing of 0.12 nm. Simulation in the $NpT$ ensemble was achieved by semi-isotropic coupling to Berendsen barostats[55] at 1 bar with compressibilities of $4.5 \times 10^{-5}$/bar and coupling constants of 4 ps; temperature-coupling was achieved using velocity Langevin dynamics[56] at 310 K with a coupling constant of 1 ps. The integration time step was 2 fs. The nonbonded pairlist was updated every 20 fs. Position-restrained equilibration periods were omitted from all statistical analyses.

### Rigidity-based allosteric communication.
To probe the allosteric transmission between the extracellular domain and the intracellular G-protein-binding region, we utilized the RTA algorithm, a computational approach based on rigidity theory[57] and an extension of the FIRST method[58]. The RTA algorithm is founded on rigidity theory, initially introduced in 2012[59,60], and further discussed by Whiteley et al.[61]. RTA predicts the extent to which a perturbation of rigidity at one binding site can be allosterically transmitted to a second distant site. Starting with an X-ray crystal structure, FIRST generates a constraint network, where the protein is modeled in terms of nodes (atoms) and edges (i.e. constraints representing covalent bonds, hydrogen bonds, electrostatic interactions, and hydrophobic contacts). Hydrogen bonds are ranked in terms of overall strength using a Mayo potential[58], whereupon a hydrogen bond energy cutoff value is selected such that all bonds weaker than this cutoff are ignored. FIRST then applies the pebble game algorithm[32,57] that rapidly decomposes a resulting network into rigid clusters and flexible regions, enabling an evaluation of non-trivial degrees of freedom (DOF) throughout the protein.

After generating the output of FIRST we applied the RTA algorithm to predict if local mechanical perturbations of rigidity at extracellular binding regions

(mimicking ligand binding) could propagate across the protein network (TM region) and lead to a quantifiable change in rigidity and conformational degrees of freedom at the intracellular G-protein-binding region, hence resulting in allosteric transmission. Equivalently, presence of rigidity-based allostery means that a change in shape (conformation) at one site could lead to rearrangement and change of shape of the second site. The number of conformational degrees of freedom at the intracellular G-protein-binding region was calculated before and after a sequential perturbation of rigidity of all residues at the extracellular region. Those residues whose rigidity perturbation induced a change in degrees of freedom (transmission of degrees of freedom through propagation of rigidity) at the intracellular region were identified as predicted allosteric sites. Residues were labeled as hot allosteric spots based on the amount of transmission of degrees of freedom.

**Rigidity transmission allostery computation**. To prepare the crystal structure for rigidity and allostery analysis, missing hydrogen atoms were added using the WHAT IF web server (http://swift.cmbi.ru.nl/servers/html/htopo.html). Starting with the N-terminus, we sequentially perturbed the rigidity of a window of three consecutive residues ($r$, $r + 1$, $r + 2$) and calculated the degrees of freedom that could be transmitted from the current window to the G-protein-binding region. Perturbation of rigidity refers to insertion of additional constraints (edges) (removal of degrees of freedom) to the window up to its rigidification. If rigidity-based allostery is present, transmission of degrees of freedom refers to a subsequent change in the available number of degrees of freedom at the G-protein-binding region. Based on the recent X-ray crystal structure of the $A_{2A}$R-G protein complex, we define residues 208, 230, and 291 located at the ends of TM5-7 respectively, to designate the G-protein-binding interface.

The RTA algorithm computes the transmission of degrees of freedom between any two sites. Denoting the current tested window as site A, and the G-protein-binding region as site B, we calculate the available conformational degrees of freedom at site A, site B and the union of sites A and B. This exercise is repeated upon successively omitting weak energy constraints in increments of 0.01 kcal/mol. Given some fixed energy cutoff, we denote these degrees of freedom counts as $A_{max}$, $B_{max}$, and $AB_{max}$, respectively (these counts were calculated using the pebble game algorithm). The maximum number of degrees of freedom that can be transmitted from A to B (denoted as DOF_AB) is finally calculated by obtaining the count DOF_AB = $A_{max}$ + $B_{max}$ − $AB_{max}$ − 6. Six is subtracted to neglect the trivial degrees of freedom corresponding to rigid body motions. When DOF_AB is positive, then sites A and B are involved in rigidity-based allosteric transmission and the maximum number of degrees of freedom that can be transmitted from A to B is DOF_AB. This provides a quantifiable measure of allosteric communication between A and B. To observe allosteric transmission for some residue $r$, a transmission curve was generated by plotting the average DOF_AB for three consecutive windows containing $r$ [i.e. ($r − 2$, $r − 1$, $r$), ($r − 1$, $r$, $r + 1$) and ($r$, $r + 1$, $r + 2$)] as a function of the energy cut-off. We calculated the intensity of allosteric transmission for each window by computing the area under the transmission curve. The intensity of allosteric transmission for residue $r$ was obtained by computing the average intensity of the three consecutive windows containing the residue $r$. Note that the intensity of allosteric transmission takes into account the number of degrees of freedom that can be transmitted and the persistence of the transmission as a function of energy strength. If allostery persists for a wide range of cutoffs, then slight changes in the hydrogen bonding network (e.g. hydrogen bond flickering) are more likely to keep the allosteric transmission active.

**HMA docking**. Docking was performed using Autodock Vina[62] version 1.1.2. Prior to docking, all non-protein structures (ions, lipids, waters) were stripped from PDB: 4EIY. PDBQT structures of HMA (5-(N, N-Hexamethylene) amiloride) and the receptor were prepared using MGLTools version 1.5.6 (AutoDockTools). All single bonds in HMA were made rotatable. A box defining the docking site was created using the following parameters: center_$x$ = −1.723, center_$y$ = −12.253, center_$z$ = 18.125, size_$x$ = 18, size_$y$ = 20, size_$z$ = 20. The lowest energy pose for the docked HMA was selected and structures were visualized in VMD 1.9.1[63].

**Data availability**. Data supporting the findings of this manuscript are available from the corresponding author upon reasonable request.

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

## Acknowledgements

This work was supported by the Natural Sciences and Engineering Research Council (NSERC) of Canada and the Canadian Institutes of Health Research (CIHR) (to R.S.P.), the Canada Excellence Research Chairs program (to O.P.E.), a postdoctoral fellowship from CIHR and US DOE LDRD funds (to C.N.), CIHR Operating Grant MOP-43998 and NSERC Discovery grant 418679 (to R.P.), NSF grant MCB-1050966 and US DOE LDRD funds (to A.E.G), National Institute of General Medical Sciences Grants RO1-GM083118, U19-GM106990 and RO1-GM068603 (R.K.S.), as well as CREST, Japan Science and Technology Agency (JST), Japan (to A.S.). Computations used resources provided by the Los Alamos National Laboratory Institutional Computing Program, which is supported by the U.S. Department of Energy National Nuclear Security Administration under Contract No. DE-AC52-06NA25396, and allocations from Compute Canada and the Extreme Science and Engineering Discovery Environment (XSEDE grant TG-MCB130178), which is funded through NSF grant ACI-1053575. O.P.E. is the Max and Anne Tanenbaum Chair in Neuroscience at University of Toronto. A Canada Foundation for Innovation (CFI) grant 19119 was used to purchase equipment used in the $^{23}Na$ and $^{25}Mg$ NMR experiments. We thank B. K. Kobilka for his suggestions and comments. We also thank A. Lai at University of Toronto for helping to set up $^{23}Na$ NMR experiments, as well V. Kanelis and J. A. Shin at University of Toronto for the use of biological instruments in their labs. We also thank A. Pandey and S. Huang for their help in preparing yeast cell pellets.

## Author contributions

L.Y., R.K.S., and R.S.P. conceived and designed the project. L.Y. performed receptor expression in *Pichia pastoris* and receptor purification. L.Y. also performed NMR labeling, performed $^{19}F$ NMR experiments, $^{23}Na$ NMR experiments, analyzed spectroscopy data, and did overall data organization and analysis for manuscript. C.N. performed molecular dynamics simulations and related data analysis. A.S. performed rigidity-transmission allostery computation and related data analysis. B.L constructed the $A_{2A}R$ baculovirus, performed *Sf9* cell expression and radioligand binding assays. D.P. performed $^{23}Na$ CPMG measurements and analysis, $^{25}Mg$ NMR, and the $Ca^{2+}$ competition assay by $^{25}Mg$ NMR. B.L. and R.K.S. analyzed and interpreted radioligand experiments. N.T. assisted with rigidity-transmission allostery computations. S.T.L. performed HMA docking experiment. A.E.G and R.P. supervised molecular dynamics simulations and related data analysis. O.P.E provided advice and revised the manuscript. L.Y. and R.S.P. prepared the manuscript. R.S.P. supervised the project.

## Additional information

**Competing interests:** The authors declare no competing interests.

