## [Peer Review File · Nature Communications]

Editorial Note: This manuscript has been previously reviewed at another journal that is not operating a transparent peer review scheme. This document only contains reviewer comments and rebuttal letters for versions considered at Nature Communications. Mentions of prior referee reports have been redacted.

Reviewers' comments:

Reviewer #1 (Remarks to the Author):

This study, coming from the experts in GPCR field, investigates role of ion binding in adenosine A2A receptor, one of the prototypical GPCRs. The interest to allosteric modulation of GPCRs by ions has been recently boosted by crystallographic discovery of Na⁺ and the corresponding highly conserved pocket in A2A and other class A GPCRs. The authors use two different NMR techniques: ²³Na probe to characterize dynamics of Na⁺ ion binding, as well as ¹⁹F probe to characterize conformational dynamics of the receptor. Then they employ MD simulations to predict potential binding sites of divalent cations and their role in positive modulation of A2A activity. The experiments provide interesting new insights into the role of the previously characterized highly conserved Na⁺ allosteric pocket in class A GPCRs, and point to a possibility of Ca²⁺ or Mg²⁺ involvement in function of A2A. The study is well written and may be of general interest to Nature communications readers.

There several issues though that have to be addressed by the authors. Most importantly, they have to clearly explain whether the observed in the study effects of divalent ion are relevant in physiological conditions. Specifically:

Line 35: "agonist, Ca²⁺ and Mg²⁺ shift the equilibrium toward the active states."

At what concentrations? Is it physiologically relevant?

Line 114: "of Ca²⁺ or Mg²⁺ shift the equilibrium of apo A2AR toward the active states, S3 and S3', as shown in Fig. 3a "

The effect shown in Fig 3 at 100mM and 500 mM concentration of Ca⁺. This is 50 times higher than physiological blood concentration of Ca⁺. It should be clearly stated and explained why such high concentration used.

Line 154: "... sodium is observed to sample the conserved binding pocket.."

What does it mean? Does this refer to nearby "secondary binding sites" as mentioned in the next section? How far they are from the Na⁺ position in the A2A inactive state structure?

Line 156: "All divalent cation binding events were observed to remain for the duration of the MD simulation"

Unclear sentence. What does it mean "observed to remain"? Remain where? in the simulation box, bound to receptor? Also, at what concentrations were the divalent ions modeled in MD simulations?

Line 200: "While the above cation-bridged interactions were observed to be long-lived by MD standards, even transient bridging of acidic residues by divalent cations would be expected to enhance the population of active or intermediate receptor states."

Again, what concentrations of divalent ions? Is it something that can be physiologically relevant?

Line 238: "While they were able to recapitulate the positive allostery on the part of Mg^{2+} , they were unable to identify a change in the allosteric response based on agonist affinity resulting from either E161A or E169Q mutations."

What about other E(D)->A mutations? were they all inconsistent with authors proposal that extracellular acidic residues involved in X2 allostery?

Line 242: "In this sense, the MD studies provide essential insight into the role of the extracellular acidic residues in facilitating divalent cation binding and contributing to the activation process."

Not clear, does it mean that only MD supports the hypothesis, but experiments cannot capture it?

Line 245 "Destabilizing the ionic lock"

This section is not about ionic lock, it rather serves as a conclusion, should be renamed "Conclusions".

Line 857: Supplementary Figure 1,b.

Lack of effect in this experiment may have at least two alternative interpretations: 1. K^+ does not bind to the Na^+ pocket 2. K^+ displaces Na^+ in the pocket, but have the same functional effect as Na^+ . Can the authors devise an experiment to distinguish between these two cases?

Reviewer #2 (Remarks to the Author):

In this current study, the research team build on their previous study published in Nature, adding additional A2aR conformational sampling data to that already published by this group. The current study focuses on the ability of cations to influence receptor conformational selection. The authors very elegantly demonstrate the ability of Na^+ to stabilise conformations associated with inactive receptor states, while Mg^{2+} and Ca^{2+} stabilise active receptor conformations that are further enhanced in the presence of agonist and G protein fragments. The paper is very well written, the experiments are very well executed, the authors conclusions are well supported by the data and the paper catalogues a number of new facts, however, the findings as they stand are mainly incremental.

For example, it has already been demonstrated numerous times that Na^+ is a negative allosteric modulator of numerous GPCRs, including the A2aR (Gao et al., 2000 Biochem Pharmacol; Gutierrez-De-Terán et al., 2013, Structure; Katritch et al., 2013; Massink et al., 2015, Mol Pharm; Kearney et al, 2014, Plos One plus others). Not only this, the mechanistic basis (at the structural level) of this modulation was revealed by mutagenesis and functional data, and then later confirmed by the publication of the 1.8Å crystal structure of the A2aR (Liu et al., 2012, Science and extensively reviewed in Katritch et al 2013). The new information provided in this study lies in the experimental observation that this binding of Na^+ shifts the conformational landscape towards the inactive conformation, an observation that is not surprising and has already been predicted and reinforced by the observations of the authors in their original Nature study that characterised these conformational states. The other novel piece of information was the transient nature of Na^+ binding to the apo receptor that may perhaps be linked to basal receptor activity.

While not as well studied, the positive modulatory effects of Mg^{2+} and Ca^{2+} have also been demonstrated previously for the A2aR (Johnsson et al., 1992 and Kim et al., 1996), although not

as elegantly as in this study. Again, given the authors previous observations on the ability of agonists and a G protein fragment to shift the proportion of receptors towards the active states (and the numerous studies on the beta 2 adrenergic receptor and other GPCRs that reveal the interplay between agonists and G protein in conformational selection), it was not unexpected that Mg²⁺ and Ca²⁺ also shifted the A2aR equilibrium towards these states (although this is the first direct demonstration of it).

The most novel part of the study lies within the molecular dynamics simulations and the application of the rigidity-transmission allostery (RTA) algorithms. This section of the study provides potential mechanistic basis for the positive allosteric effects observed from these cations. However, while the findings of these studies are certainly very interesting, they lack further experimental validation. This would need to be provided to enhance the impact of these findings. In particular, findings of J Kim et al, 1996 (Mol Pharmacol) do not experimentally support a role for glutamic acid residues in ECL2 of A2aR in the binding of Ca²⁺ and Mg²⁺ (major interactions in the bridging mechanism proposed by the authors as one of the methods of positive modulation by these cations). The authors attempted to address this suggesting this was due to the loss of agonist affinity by receptor mutations, however, the study still revealed no loss of cation positive cooperativity by the mutations. In light of this, at least some part of the proposed mechanism(s) from the in silico studies will need to be demonstrated experimentally.

In addition, the abstract clearly states that an understanding of cation allostery should enable the design of novel allosteric agents. To do this would enable a clear mechanistic understanding of how this arises that is currently not provided in the study as it stands.

Minor Comments:

The introduction mentions the observations of positive allostery of Mg²⁺ and Ca²⁺ for GPCRs, but does not include A2aR that has been revealed previously. This should be added.

The application of RTA algorithms is first described in the discussion. Some of this section belongs in the results, not the discussion (ie lines 206-224).

The G alpha S peptide fragment size and sequence is not described in the methods.

Labelling of Figure 4d is incorrect. Legend is same as for C. This should be Ca²⁺ not Mg²⁺??

Reviewer #3 (Remarks to the Author):

This paper describes the impact of Na⁺ and of divalent (Ca²⁺ and Mg²⁺) on the conformational equilibria of a G protein coupled receptor (the A2 adenosine receptor) and the allosteric coupling of cation binding to binding of other ligands. There is truly excellent experimental and computational data in this paper. I particularly salute the authors for working out experimental conditions for the NMR studies of this work.

However, I have some major concerns about this work that I think need to be addressed before this work should be published in Nature Communications. Most of the NMR and the computational studies of this work were conducted at high (100 mM and higher) cation concentrations. Moreover, the protein-micelle (experimental) and protein-membrane (computational) were subjected to uniform ion concentrations. This is problematic in two ways. First, while extracellular Na⁺ does exceed 100 mM, neither intra- nor extra-cellular free divalent ion concentrations are likely to exceed low mM under physiological conditions. Thus, some of the key results of this work were

carried out at ion concentrations much higher (sometimes by orders of magnitude) than maximal physiological concentrations. Moreover, the intra- versus extracellular concentrations of different ions are highly compartmentalized under cellular conditions. Extracellular Na^+ exceed 100 mM, but intracellular levels are more on the order of 10 mM. Intracellular Ca^{2+} is micromolar (extracellular around 1 millimolar), while intracellular free Mg^{2+} is roughly 1 millimolar. The authors seem to have made little effort to try to match experimental ion concentrations to physiological concentrations or to take into account that ion binding sites on the extracellular face will experience very different ion concentrations under physiological conditions than sites located on the intracellular face. I note also that the authors did do a cellular experiment (lines 130-135, Fig. 4) where they found that extracellular Mg^{2+} and Ca^{2+} activated binding of another ligand with Kact in the sub-millimolar range—two orders of magnitude lower than the concentrations used in their NMR experiments. One wonder if the changes divalent cation-induced changes seen at 100-500 mM by NMR have any relationship to the activation of function observed in cell experiments where Kact is sub-millimolar for both Ca^{2+} and Mg^{2+} .

At the very least, the authors should constantly be reminding the readers of the often highly supra-physiological concentrations of ions used in their NMR studies and also should frequently remind readers of the intra- versus extracellular ion concentration dilemma. When describing binding sites they also should usually state which face of the receptor the site is on.

Another concern with this work is the while molecular dynamics simulations (at high uniform ion concentrations) suggest the location of ion binding sites, determining which of these sites are physiologically relevant is not determined in this work. While all mutagenesis studies have their perils, I do think most readers would expect to see attempts made to confirm which computationally-predicted sites are responsible for the cation-induced changes in equilibria that are documented by NMR or receptor ligand binding/functional measurements.

I view this paper as a glass half full work and encourage the authors to revise their manuscript so that the quality of their data is matched by the sophistication of how they relate their results to physiological conditions.

Our specific responses to the comments by the reviewers are found below.

Reviewers' comments:

Reviewer #1 (Remarks to the Author):

This study, coming from the experts in GPCR field, investigates role of ion binding in adenosine A2A receptor, one of the prototypical GPCRs. The interest to allosteric modulation of GPCRs by ions has been recently boosted by crystallographic discovery of Na⁺ and the corresponding highly conserved pocket in A2A and other class A GPCRs. The authors use two different NMR techniques: ²³Na probe to characterize dynamics of Na⁺ ion binding, as well as ¹⁹F probe to characterize conformational dynamics of the receptor. Then they employ MD simulations to predict potential binding sites of divalent cations and their role in positive modulation of A2A activity.

The experiments provide interesting new insights into the role of the previously characterized highly conserved Na⁺ allosteric pocket in class A GPCRs, and point to a possibility of Ca²⁺ or Mg²⁺ involvement in function of A2A. The study is well written and may be of general interest to Nature communications readers.

There several issues though that have to be addressed by the authors. Most importantly, they have to clearly explain whether the observed in the study effects of divalent ion are relevant in physiological conditions. Specifically:

Line 35: "agonist, Ca²⁺ and Mg²⁺ shift the equilibrium toward the active states."

At what concentrations? Is it physiologically relevant?

Line 114: "of Ca²⁺ or Mg²⁺ shift the equilibrium of apo A2AR toward the active states, S₃ and S₃', as shown in Fig. 3a "

The effect shown in Fig 3 at 100mM and 500 mM concentration of Ca⁺. This is 50 times higher than physiological blood concentration of Ca⁺. It should be clearly stated and explained why such high concentration used.

We thank the reviewer for his/her comments. We have repeated the ¹⁹F NMR measurements in MNG micelles at lower cation concentrations. We observe distinguishable effects on the population of the active state referred to as S₃ at 20 mM Mg²⁺ (See supplementary Figure. 6) in the presence of a Gαs peptide (the C-terminal peptide of the G-protein known to insert into the active state cytosolic pocket of the receptor). In contrast, an enhancement in NECA binding can be seen *via* radioligand binding assays in insect cell membranes at 350 μM cation concentrations with much lower concentration of receptor used in comparison to NMR experiments, and using full Gαs protein. Thus, it may be that the interaction with the full Gαs-protein and/or the presence of lipids is needed to achieve efficient allosteric effects with Mg²⁺ at physiological concentrations or that detergent weakens this allosteric interaction. We nevertheless believe that qualitatively the NMR and radioligand experiments are recapitulating the same allosteric effect. There are also some compelling parallels between the radioligand data and the ¹⁹F NMR data: 1) Positive allosteric effects are observed with Ca²⁺ and Mg²⁺ but not Fe²⁺ using the radioligand assay. This is exactly the case in the NMR experiments. The recent NMR experiments with Fe²⁺ are now discussed in the revised manuscript and presented in Supplementary Figure 9, 2) Positive allosteric effects are only observed when Ca²⁺ or Mg²⁺ are combined with full agonist rather than partial agonist for both radioligand binding assays and ¹⁹F NMR. We hope to present this as part of a later paper on partial agonism. Finally,

spectroscopic studies of A_{2A}R in micelles reliably recapitulate functional states which we have recently validated by DEER spectroscopy. However, the ligand concentrations used to obtain saturable effects are often much higher than those used in cell assays. In general, micelles qualitatively reproduce responses observed in cells although we view the receptor as adopting a more dynamic and possibly “looser” structure in which case it is not surprising that we must add more ligand or cation to recapitulate what is seen in cells.

Line 154: “... sodium is observed to sample the conserved binding pocket..”

What does it mean? Does this refer to nearby “secondary binding sites” as mentioned in the next section? How far they are from the Na⁺ position in the A_{2A} inactive state structure?

We were referring to excursions of sodium in the active state structure of A_{2A} based on MD simulations. We have since completely rewritten this section and added more quantitative data regarding sodium exchange. The main conclusions are: 1) sodium exchanges with its conserved binding site pocket identified by X-ray crystallography, which we corroborate by a competition experiment with amiloride (Fig.2c and d). ²³Na CPMG measurements obtained at multiple magnetic field strengths show compelling evidence for an average bound state lifetime of sodium that is around 480 μs and which incidentally changes upon adding inverse agonist (Fig.2e), and 2) The activation intermediate state, S₃, which is stabilized by partial agonist, is also capable of binding sodium via an independent and likely weaker sodium binding site. While this is part of recent unpublished work, we have validated this by additional ²³Na CPMG experiments in the presence of agonist, where we see evidence for a second weaker sodium binding site on the receptor. The main point, which is discussed in the revised manuscript, is that we can now begin to understand how sodium egress may result in the activation process.

Line 156: “All divalent cation binding events were observed to remain for the duration of the MD simulation”

Unclear sentence. What does it mean “observed to remain”? Remain where? in the simulation box, bound to receptor? Also, at what concentrations were the divalent ions modeled in MD simulations?

We have rewritten this sentence to clarify “All of the divalent cation-mediated interactions described below persisted for the duration of the simulation and are classified in terms of...” (Page 8, lines 175-176).

Line 200: “While the above cation-bridged interactions were observed to be long-lived by MD standards, even transient bridging of acidic residues by divalent cations would be expected to enhance the population of active or intermediate receptor states.”

Again, what concentrations of divalent ions? Is it something that can be physiologically relevant?

We employed high divalent cation starting concentrations to explore what regions of the receptor divalent cations might interact with. We then explored how such binding might allosterically modulate activation using a mathematical approach called rigidity transmission allostery. Indeed, we did see positive allostery from divalent cations effectively compacting the extracellular domains through bridging interactions but we argue that this may well NOT be physiologically relevant, given the high divalent cation concentrations used in the simulations

(effectively, 300 mM). However, we also argue that this is a sensible computational tool for exploring how specific allosteric interactions can influence activation as a means to design drugs which bind the same regions and achieve the same end. On the other hand, we note that there is a well-known physiologically relevant positive allosteric effect of Ca^{2+} and Mg^{2+} in $\text{A}_{2\text{A}}\text{R}$ that likely arises from a conserved binding site, yet to be identified. These effects are observed at 350 μM cation in our radioligand measurements albeit at lower receptor concentration than that used in NMR experiments (20 mM). As mentioned above, the positive allosterism is exclusive to Mg^{2+} and Ca^{2+} and not Fe^{2+} which we observe spectroscopically and via radioligand studies.

Line 238: “While they were able to recapitulate the positive allosterism on the part of Mg^{2+} , they were unable to identify a change in the allosteric response based on agonist affinity resulting from either E161A or E169Q mutations.”

What about other E(D)->A mutations? were they all inconsistent with authors proposal that extracellular acidic residues involved in X2 allosterism?

Our point was that most of the mutations of the acidic residues significantly curtailed ligand binding and likely many of these residues play a key role in ligand selectivity. Thus testing by mutation for loss of positive allosterism is difficult. In our case we identified divalent cation mediated interactions associated with D170-D261, E151-E161, E151-E169-D170, E169-D170. E169 and D170 are also likely interconvertible making the prospect of single mutations difficult. Moreover, it is also possible that the proposed allosteric effects resulting from bridging key acidic residues, arise from multiple transient and cooperative interactions, as proposed in Fig. 5. (Please also see detailed discussions on Page 14, lines 301-314). In the revised manuscript, we emphasize the mechanistic discussion of sodium interaction dynamics and simply make the claim that at high concentrations of divalent cation, we observe these interactions by MD simulations and we can make a connection to an allosteric response by rigidity analysis.

Line 242: “In this sense, the MD studies provide essential insight into the role of the extracellular acidic residues in facilitating divalent cation binding and contributing to the activation process.” Not clear, does it mean that only MD supports the hypothesis, but experiments cannot capture it?

We were trying to say that the role of the acidic residues is likely two-fold – ligand selectivity and allosterism - making it difficult to perform mutations aimed at understanding allosterism, while straightforward to perform MD simulations. Please see the corresponding discussion (Page 14, lines 301-314).

Line 245 “Destabilizing the ionic lock”

This section is not about ionic lock, it rather serves as a conclusion, should be renamed “Conclusions”.

This section has been moved.

Line 857: Supplementary Figure 1,b.

Lack of effect in this experiment may have at least two alternative interpretations: 1. K^+ does not bind to the Na^+ pocket 2. K^+ displaces Na^+ in the pocket, but have the same functional effect as Na^+ . Can the authors devise an experiment to distinguish between these two cases?

This is an interesting question. As shown in Supplementary Fig.1, there is no significant effect of K^+ over a wide range of concentrations attempted. Therefore, for the present paper, we feel the results are compelling – namely that titrating potassium alone has no significant effect on the

population of the observable states (S_{12} , S_3 , and S_3') implying no stabilizing interaction within the accuracy of the experiment. Obviously, sodium titrations show significant responses. Similar results were also reported recently by Christopher Tate's group, please see Extended Data Figure 1 in their paper. It is clear that K^+ does not exert significant stabilizing effects with regard to inverse agonist ZM241385 binding, while sodium dramatically enhances ZM binding affinity (Carpenter et al., Nature, 2016).

Reviewer #2 (Remarks to the Author):

In this current study, the research team build on their previous study published in Nature, adding additional A2aR conformational sampling data to that already published by this group. The current study focuses on the ability of cations to influence receptor conformational selection. The authors very elegantly demonstrate the ability of Na^+ to stabilise conformations associated with inactive receptor states, while Mg^{2+} and Ca^{2+} stabilise active receptor conformations that are further enhanced in the presence of agonist and G protein fragments. The paper is very well written, the experiments are very well executed, the authors conclusions are well supported by the data and the paper catalogues a number of new facts, however, the findings as they stand are mainly incremental.

For example, it has already been demonstrated numerous times that Na^+ is a negative allosteric modulator of numerous GPCRs, including the A2aR (Gao et al., 2000 Biochem Pharmacol; Gutierrez-De-Terán et al., 2013, Structure; Katritch et al., 2013; Massink et al., 2015, Mol Pharm; Kearney et al, 2014, Plos One plus others). Not only this, the mechanistic basis (at the structural level) of this modulation was revealed by mutagenesis and functional data, and then later confirmed by the publication of the 1.8Å crystal structure of the A2aR (Liu et al., 2012, Science and extensively reviewed in Katritch et al 2013). The new information provided in this study lies in the experimental observation that this binding of Na^+ shifts the conformational landscape towards the inactive conformation, an observation that is not surprising and has already been predicted and reinforced by the observations of the authors in their original Nature study that characterised these conformational states. The other novel piece of information was the transient nature of Na^+ binding to the apo receptor that may perhaps be linked to basal receptor activity.

While not as well studied, the positive modulatory effects of Mg^{2+} and Ca^{2+} have also been demonstrated previously for the A2aR (Johnsson et al., 1992 and Kim et al., 1996), although not as elegantly as in this study. Again, given the authors previous observations on the ability of agonists and a G protein fragment to shift the proportion of receptors towards the active states (and the numerous studies on the beta 2 adrenergic receptor and other GPCRs that reveal the interplay between agonists and G protein in conformational selection), it was not unexpected that Mg^{2+} and Ca^{2+} also shifted the A2aR equilibrium towards these states (although this is the first direct demonstration of it).

We thank the reviewer for his/her comments. We have changed the content and tone of the original manuscript considerably. If NMR could only recapitulate an inactive and an active state, we would agree that the observation that sodium (a known NAM) tips the balance toward inactive is "incremental". However, NMR reveals a more complex world, consisting in this case of the inactive ensemble (S_{12}) along with two distinct conformers corresponding to two unique active-like states, S_3 and S_3' . In particular, the activation intermediate, known to be stabilized by partial agonist, is also promoted by sodium suggesting that sodium is able to bind to both inactive and at least one activation intermediate state (conformer). We also put this on a firmer

footing by making definitive and quantitative measurements of sodium interaction kinetics. We feel this methodology (*i.e.* metal NMR CPMG relaxation dispersions) could find fantastic new applications in the study of metal interaction dynamics with receptors.

The most novel part of the study lies within the molecular dynamics simulations and the application of the rigidity-transmission allostery (RTA) algorithms. This section of the study provides potential mechanistic basis for the positive allosteric effects observed from these cations. However, while the findings of these studies are certainly very interesting, they lack further experimental validation. This would need to be provided to enhance the impact of these findings. In particular, findings of J Kim et al, 1996 (Mol Pharmacol) do not experimentally support a role for glutamic acid residues in ECL2 of A2aR in the binding of Ca²⁺ and Mg²⁺ (major interactions in the bridging mechanism proposed by the authors as one of the methods of positive modulation by these cations). The authors attempted to address this suggesting this was due to the loss of agonist affinity by receptor mutations, however, the study still revealed no loss of cation positive cooperativity by the mutations. In light of this, at least some part of the proposed mechanism(s) from the *in silico* studies will need to be demonstrated experimentally.

We have retained the discussion of the MD simulations and Rigidity Transmission Allostery theory as useful methodological tools. As discussed in our response to reviewer 1, we performed additional studies to suggest there is some conserved binding site, cooperative with agonist, responsible for positive allostery although we can't yet point to where it is. We make the point that the compression of the extracellular domain by salts or other means is a valid allosteric tool that has been used recently on the muscarinic receptor. The bottom line however is that the divalent ion mediated bridging of acidic residues may not be physiological.

In addition, the abstract clearly states that an understanding of cation allostery should enable the design of novel allosteric agents. To do this would enable a clear mechanistic understanding of how this arises that is currently not provided in the study as it stands.

The new quantitative measurements with sodium and the interaction dynamics and consequences to signaling etc are hopefully what the reviewer was looking for.

Minor Comments:

The introduction mentions the observations of positive allostery of Mg²⁺ and Ca²⁺ for GPCRs, but does not include A2aR that has been revealed previously. This should be added.

Thank you. Added accordingly (Page 3, lines 57-59)

The application of RTA algorithms is first described in the discussion. Some of this section belongs in the results, not the discussion (*ie* lines 206-224).

We have tried to rebalance the results and discussion sections regarding RTA

The G alpha S peptide fragment size and sequence is not described in the methods.

Discussed and referenced in Reagents and Biochemical Assay. (Page 21, line 468)

Labelling of Figure 4d is incorrect. Legend is same as for C. This should be Ca²⁺ not Mg²⁺??

Thank you. Corrected accordingly.

Reviewer #3 (Remarks to the Author):

This paper describes the impact of Na⁺ and of divalent (Ca²⁺ and Mg²⁺) on the conformational equilibria of a G protein coupled receptor (the A2 adenosine receptor) and the allosteric coupling of cation binding to binding of other ligands. There is truly excellent experimental and computational data in this paper. I particularly salute the authors for working out experimental conditions for the NMR studies of this work.

However, I have some major concerns about this work that I think need to be addressed before this work should be published in Nature Communications. Most of the NMR and the computational studies of this work were conducted at high (100 mM and higher) cation concentrations. Moreover, the protein-micelle (experimental) and protein-membrane (computational) were subjected to uniform ion concentrations. This is problematic in two ways. First, while extracellular Na⁺ does exceed 100 mM, neither intra- nor extra-cellular free divalent ion concentrations are likely to exceed low mM under physiological conditions. Thus, some of the key results of this work were carried out at ion concentrations much higher (sometimes by orders of magnitude) than maximal physiological concentrations. Moreover, the intra- versus extracellular concentrations of different ions are highly compartmentalized under cellular conditions. Extracellular Na⁺ exceed 100 mM, but intracellular levels are more on the order of 10 mM. Intracellular Ca²⁺ is micromolar (extracellular around 1 millimolar), while intracellular free Mg²⁺ is roughly 1 millimolar. The authors seem to have made little effort to try to match experimental ion concentrations to physiological concentrations or to take into account that ion binding sites on the extracellular face will experience very different ion concentrations under physiological conditions than sites located on the intracellular face. I note also that the authors did do a cellular experiment (lines 130-135, Fig. 4) where they found that extracellular Mg²⁺ and Ca²⁺ activated binding of another ligand with Kact in the sub-millimolar range—two orders of magnitude lower than the concentrations used in their NMR experiments. One wonder if the changes divalent cation-induced changes seen at 100-500 mM by NMR have any relationship to the activation of function observed in cell experiments where Kact is sub-millimolar for both Ca²⁺ and Mg²⁺.

At the very least, the authors should constantly be reminding the readers of the often highly supra-physiological concentrations of ions used in their NMR studies and also should frequently remind readers of the intra- versus extracellular ion concentration dilemma. When describing binding sites they also should usually state which face of the receptor the site is on.

These are all excellent points which were also echoed by reviewers above. As the reviewer recognized, physiological extracellular concentrations of sodium chloride are on the order of 140 mM while divalent cation concentrations would rarely be more than low mM. In the former case, we observe that sodium exhibits a dissociation constant that is on the order of 60 mM to the apo receptor (roughly half that of physiological concentration). Moreover, we observe bound state life times of sodium to be on average, 500 μs for sodium. The relatively weak affinity of sodium and rapid interaction dynamics are both key to the role which sodium plays both as a NAM and at the same time a sensitive switch in the activation process. As discussed above, our new CPMG relaxation dispersion measurements performed on the apo receptor and the inverse agonist-stabilized receptor, allow us to elaborate on basal activity and inverse agonism. More importantly, we hope that readers will appreciate the utility of these measurements with regard to the study of interaction dynamics of allosteric cations with receptors, using metal NMR CPMG relaxation dispersion measurements. The reviewer quite rightly critiqued our work with regard to the high concentrations of divalent cation used in our NMR experiments. We recognize this but

at the same time, discuss the notion of compression of the extracellular domain (in this case, via divalent cation mediated bridges) of a GPCR as a means of allosterically stabilizing the active state. The MD and computational studies are presented as tools for validating such ideas. We then focus on the interesting parallels between the radioligand studies and the NMR studies with divalent cation as discussed above (*i.e.* **1**) both studies see positive allosteric effects only with full agonist and G protein or G protein peptide, **2**) both studies see positive allostery only with Mg^{2+} and Ca^{2+} and not Fe^{2+}). We discuss the limitations of the current NMR studies in micelles in terms of recapitulating positive allostery, albeit at higher divalent cation concentrations (20 mM) possibly because the G-protein surrogate was a peptide in the NMR studies or possibly because of the fallibility of micelles versus cell membranes.

In short, we fully recognize that the divalent cation concentrations used (as low as 20 mM) where we see allosteric effects by NMR are still over an order of magnitude higher than that observed on the same construct in cells. We discuss complexities associated with the micelle and the G protein peptide. While we do suggest there is likely a divalent cation-specific binding site, we don't know if cation-mediated cross linking of acidic residues is physiologically relevant.

Another concern with this work is the while molecular dynamics simulations (at high uniform ion concentrations) suggest the location of ion binding sites, determining which of these sites are physiologically relevant is not determined in this work. While all mutagenesis studies have their perils, I do think most readers would expect to see attempts made to confirm which computationally-predicted sites are responsible for the cation-induced changes in equilibria that are documented by NMR or receptor ligand binding/functional measurements.

Agreed. As discussed, we have refocused our discussion from a mechanistic basis, on sodium allostery and interaction dynamics. The observation of additional binding sites and quantitation of interaction kinetics by the ^{23}Na NMR were previously not known and have allowed us to speak more about sodium egress, basal activation and mechanisms of activation.

I view this paper as a glass half full work and encourage the authors to revise their manuscript so that the quality of their data is matched by the sophistication of how they relate their results to physiological conditions.

We thank the reviewer for his/her criticisms or opinions. We think that the combined NMR and radioligand studies have brought additional clarity to the positive allosteric effects with divalent cations but more importantly, we have flushed out a compelling story of negative allostery by sodium and the ensuing interaction dynamics.

Reviewers' Comments:

Reviewer #1:

Remarks to the Author:

The authors addressed all major concerns and improved the paper by adding new CPMG data.

However, there are still a few minor items below to be corrected before the paper can be accepted to Nat Comm:

Abstract, Line 40: "Molecular dynamics simulations reveal that"

"Reveal" should be replaced with "suggest" or "predict". MD just cannot "reveal".

Discussion, Line 214: "suggestive of a lower affinity Na⁺ binding site"

This is an important new statement, also repeated in line 216 "clear affinity to A2AR", but it does not make any sense without at least some quantitative estimation of this affinity. Apparently, Na⁺ has SOME affinity to any GPCR or any protein that has acidic residues exposed.

Reference duplication: 22. Katritch et al TiBS 2014, also repeated as ref.38

Reviewer #2:

Remarks to the Author:

The revised version of the manuscript is considerably improved relative to the initial submission, addressing the majority of my concerns. However, while the authors have made some attempts to address the mechanism of the observed allostery by demonstrating additional cation binding sites exist and through quantification of the interaction kinetics, there is still little attempt to experimentally demonstrate the location of the cation binding sites identified in the MD. The authors address this by stating that these effects are impossible to identify via mutagenesis, but it is still unclear why this is so and it is not clear if the authors made any attempt to actually do this in their own lab, either by using functional or binding assays or through doing NMR based experiments on mutant receptors. In addition, the authors speculate that while these sites identified in the MD may not be physiological, compression of the extracellular domain by salts is a valid means to allosterically target these receptors. To state this still requires experimental validation of the sites identified in the MD simulations.

Reviewer #3:

Remarks to the Author:

The authors have greatly improved this paper in response to reviewer concerns. However, some of the fundamental concerns regarding the supraphysiological concentrations of Ca(II) and Mg(II) examined in their NMR experiments and computations persist. Understandably, the authors chose to address these concerns with text rather than repeating all experiments/calculations. The one additional change in the text that I think is needed is that the authors should explicitly state at the bottom of page 6 and top of page 7 of the Results section the Mg(II) and Ca(II) concentrations used in their NMR experiments.

Despite thees lingering concerns I come down in favor of publication, both because the technical novelty of their nice ²³Na NMR experiments and because the insight this paper provides into metal ion modulation of GPCRs.

REVIEWERS' COMMENTS:

Reviewer #1 (Remarks to the Author):

The authors addressed all major concerns and improved the paper by adding new CPMG data. However, there are still a few minor items below to be corrected before the paper can be accepted to Nat Comm:

Abstract, Line 40: “Molecular dynamics simulations reveal that”
“Reveal” should be replaced with “suggest” or “predict”. MD just cannot “reveal”.

Done (Page 2, line 39)

Discussion, Line 214: “suggestive of a lower affinity Na⁺ binding site”. This is an important new statement, also repeated in line 216 “clear affinity to A2AR”, but it does not make any sense without at least some quantitative estimation of this affinity. Apparently, Na⁺ has SOME affinity to any GPCR or any protein that has acidic residues exposed.

In control experiments with albumin, we observe a very weak interaction between sodium and the protein and no interactions with lipids or micelles. Thus it isn't as easy as it sounds to see evidence for binding. Acidic residues alone shouldn't be sufficient to detect a bound sodium site. Moreover, the interaction is clearly modulated by drug. We do believe there are at least two binding sites as discussed. The affinity should not be more than an order of magnitude different than that associated with the primary binding site. Details of the sodium interaction network are a year away but we wanted to point the reader to the possibilities of metal NMR. (Page 10, lines 215-218)

Reference duplication: 22. Katritch et al TiBS 2014, also repeated as ref.38

Thank you very much! We removed the ref.38.

Reviewer #2 (Remarks to the Author):

The revised version of the manuscript is considerably improved relative to the initial submission, addressing the majority of my concerns. However, while the authors have made some attempts to address the mechanism of the observed allostery by demonstrating additional cation binding sites exist and through quantification of the interaction kinetics, there is still little attempt to experimentally demonstrate the location

of the cation binding sites identified in the MD. The authors address this by stating that these effects are impossible to identify via mutagenesis, but it is still unclear why this is so and it is not clear if the authors made any attempt to actually do this in their own lab, either by using functional or binding assays or through doing NMR based experiments on mutant receptors. In addition, the authors speculate that while these sites identified in the MD may not be physiological, compression of the extracellular domain by salts is a valid means to allosterically target these receptors. To state this still requires experimental validation of the sites identified in the MD simulations.

We presume the reviewer is referring specifically to experimentally validating the location of cation binding sites associated with divalent cations (i.e. the latter part of our story). In a seminal paper by Jacobson and colleagues, acidic residues in the second extracellular loop (E151, E161, E169, and D170) were replaced with alanine or glutamine residues as discussed in the paper. High affinity agonist and antagonist binding was not observed for E151A, E151Q, E151D, and E169A although E151A and E169A showed full stimulation of adenylyl cyclase. D170K behaved like wild type. In general, their main conclusion was that the acidic residues appeared to be directly involved in ligand binding. We did make an E151A mutation and we observed no change in positive allostery (at low $[Ca^{2+}]$ or low $[Mg^{2+}]$) by a radioligand binding assay, confirming the earlier work of Jacobson. However, our hesitation with regard to embarking on an exhaustive mutational analysis is: 1) ligand affinity is clearly affected. This complicates analysis of allosteric effects through radioligand measurements, 2) The effects we observed through MD simulations were a result of very high concentrations of divalent cation. Thus, the contributions to allostery from bridging of acidic residues may be cooperative and thus difficult to parse without additional double/multiple mutants by silencing double/multiple cation bridges, 3) The bulk of the mutational work has been done by Jacobson, resulting in ambiguous results regarding allostery, and 4) We could make the above mutants from the perspective of studying changes in conformational equilibria using NMR, although this would represent a monumental effort. In our hands, each mutant requires 3-6 months of optimization. This is because once we introduce a new mutant in yeast (*Pichia*), we need to re-optimize expression levels on increasing amounts of geneticin to increase the copy number of the A2A gene in *Pichia*. NMR requires significant amounts of material and thus, significant effort to reach requisite expression levels.

With regard to compression, the paper presented the results of MD simulations and the conformational changes that ensue in the presence of divalent cations over 1 μ s. As the reviewer wrote, we suggest that this may not be physiological but nonetheless results in allosteric enhancement of activation. We can validate this positive allostery through Rigidity Transmission Allostery which utilizes the active state structures and evaluates the consequences of such cross-links to allosteric opening of the G protein binding domain. The compaction of the extracellular domains by divalent cations is validated by comparison with published crystal structures, as summarized in Figure 8. Specifically, our MD simulations point toward the coming together of D261 and D170. The vast majority of active state structures of A2A show a similar trend (Figure 8b). In addition, for these structures, we see a corresponding opening of the G protein binding cavity

(Figure 8e-f). Thus, the notion of compaction of the extracellular region and corresponding expansion of the intracellular region is something quite familiar to the activation process. Finally, we note that compaction of the extracellular domain as an allosteric mechanism for activation has been successfully demonstrated in the M2 muscarinic acetylcholine receptor in which the allosteric modulator, LY2119620, achieved exactly this compaction.

Reviewer #3 (Remarks to the Author):

The authors have greatly improved this paper in response to reviewer concerns. However, some of the fundamental concerns regarding the supraphysiological concentrations of Ca(II) and Mg(II) examined in their NMR experiments and computations persist. Understandably, the authors chose to address these concerns with text rather than repeating all experiments/calculations. The one additional change in the text that I think is needed is that the authors should explicitly state at the bottom of page 6 and top of page 7 of the Results section the Mg(II) and Ca(II) concentrations used in their NMR experiments.

Done (at the bottom of Page 6)

Despite these lingering concerns I come down in favor of publication, both because the technical novelty of their nice ^{23}Na NMR experiments and because the insight this paper provides into metal ion modulation of GPCRs.

Thanks!